# Simplicity Bias in 1-Hidden Layer Neural Networks

**Depen Morwani** *
Department of Computer Science
Harvard University
dmorwani@g.harvard.edu

**Jatin Batra**
School of Technology and Computer Science
Tata Institute of Fundamental Research (TIFR)
jatin.batra@tifr.res.in

**Prateek Jain**[1]
Google Research
prajain@google.com

**Praneeth Netrapalli**[1]
Google Research
pnetrapalli@google.com

## Abstract

Recent works (Shah et al., 2020; Chen et al., 2021) have demonstrated that neural networks exhibit extreme *simplicity bias* (SB). That is, they learn *only the simplest* features to solve a task at hand, even in the presence of other, more robust but more complex features. Due to the lack of a general and rigorous definition of *features*, these works showcase SB on *semi-synthetic* datasets such as Color-MNIST, MNIST-CIFAR where defining features is relatively easier.

In this work, we rigorously define as well as thoroughly establish SB for *one hidden layer* neural networks. More concretely, (i) we define SB as the network essentially being a function of a low dimensional projection of the inputs (ii) theoretically, in the infinite width regime, we show that when the data is linearly separable, the network primarily depends on only the linearly separable (1-dimensional) subspace even in the presence of an arbitrarily large number of other, more complex features which could have led to a significantly more robust classifier, (iii) empirically, we show that models trained on *real* datasets such as Imagenet and Waterbirds-Landbirds indeed depend on a low dimensional projection of the inputs, thereby demonstrating SB on these datasets, iv) finally, we present a natural ensemble approach that encourages diversity in models by training successive models on features not used by earlier models, and demonstrate that it yields models that are significantly more robust to Gaussian noise.

## 1 Introduction

It is well known that neural networks (NNs) are vulnerable to distribution shifts as well as to adversarial examples (Szegedy et al., 2014; Hendrycks et al., 2021). A recent line of work (Geirhos et al., 2018; Shah et al., 2020; Geirhos et al., 2020) proposes that *Simplicity Bias (SB)* (or shortcut learning) i.e., the tendency of neural networks (NNs) to learn only the simplest features over other useful but more complex features, is a key reason behind non-robustness of the trained networks. The argument is roughly as follows: for example, in the classification of swans vs bears, as illustrated in Figure 1, there are many features such as background, color of the animal, shape of the animal etc. that can be used for classification. However using only one or few of them can lead to models that are not robust to specific distribution shifts, while using all the features can lead to more robust models.

Several recent works have demonstrated SB on a variety of *semi-real constructed datasets* (Geirhos et al., 2018; Shah et al., 2020; Chen et al., 2021), and have hypothesized SB to be the key reason

---

*part of the work done while at Google Research, India [1]Alphabetical Ordering
[2]Image source: Wikipedia swa, bea.

for NN's brittleness to distribution shifts (Shah et al., 2020). However, such observations are still only for specific semi-real datasets, and a general method that can identify SB on a *given dataset* and a *given model* is still missing in literature. Such a method would be useful not only to estimate the robustness of a model but could also help in designing more robust models.

A key challenge in designing such a general method to identify (and potentially fix) SB is that the notion of *feature* itself is vague and lacks a rigorous definition. Existing works Geirhos et al. (2018); Shah et al. (2020); Chen et al. (2021) avoid this challenge of vague feature definition by using carefully designed datasets (e.g., concatenation of MNIST images and CIFAR images), where certain high level features (e.g., MNIST features and CIFAR features, shape and texture features) are already baked in the dataset definition, and arguing about their *simplicity* is intuitively easy.

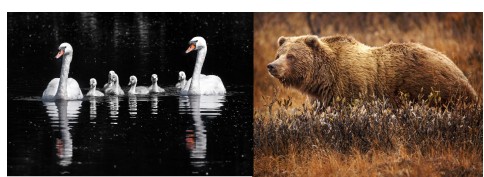

Figure 1: Classification of swans vs bears. There are several features such as background, color of the animal, shape of the animal etc., each of which is sufficient for classification but using all of them will lead to a more robust model. [2]

**Contributions**: Our first contribution is to provide a precise definition of a particular simplicity bias – LD-SB– referring to *low dimensional input dependence* of the model.

**Definition 1.1** (LD-SB). A model $f : \mathbb{R}^d \to \mathbb{R}^c$ with inputs $x \in \mathbb{R}^d$ and outputs $f(x) \in \mathbb{R}^c$ (e.g., logits for $c$ classes), trained on a distribution $(x, y) \sim \mathcal{D}$ satisfies LD-SB if there exists a *projection* matrix $P \in \mathbb{R}^{d \times d}$ satisfying:

- $\text{rank}(P) = k \ll d$,

- $\mathbb{P}[\text{pred}(f(Px^{(1)} + P_\perp x^{(2)})) = \text{pred}(f(x^{(1)}))] \geq 1 - \epsilon_1$ for $(x^{(1)}, y^{(1)}), (x^{(2)}, y^{(2)}) \sim \mathcal{D}$, where $\text{pred}(f(x))$ represents the predicted label for $x$,

- An independent model $g$ trained on $(P_\perp x, y)$ where $(x, y) \sim \mathcal{D}$ satisfies $|\text{Acc}(g) - \text{Acc}(f)| \leq \epsilon_2$,

for some small $\epsilon_1$ and $\epsilon_2$. Here $P_\perp$ is the projection matrix onto the subspace orthogonal to $P$, and $\text{Acc}(f)$ represents the accuracy of $f$.

In words, LD-SB says that there exists a small $k$-dimensional subspace (given by the projection matrix $P$) in the input space $\mathbb{R}^d$, which is the only thing that the model $f$ considers in labeling any input point $x$. In particular, if we *mix* two data points $x_1$ and $x_2$ by using the projection of $x_1$ onto $P$ and the projection of $x_2$ onto the orthogonal subspace $P_\perp$, the output of $f$ on this *mixed point* $Px_1 + P_\perp x_2$ is the same as that on $x_1$. This would have been fine if the subspace $P_\perp$ does not contain any feature useful for classification. However, the third bullet point says that $P_\perp$ indeed contains features that are useful for classification since an independent model $g$ trained on $(P_\perp x, y)$ achieves high accuracy.

Theoretically, we demonstrate LD-SB of $1$-*hidden layer NNs in the infinite width limit* for a fairly general class of distributions called *independent features model (IFM)*, where the features (i.e., coordinates) are distributed independently conditioned on the label. IFM has a long history and is widely studied, especially in the context of naive-Bayes classifiers Lewis (1998). For IFM, we show that as long as there is even a *single* feature in which the data is linearly separable, NNs trained using SGD will learn models that rely almost exclusively on this linearly separable feature, even when there are an *arbitrarily large number* of features in which the data is separable but with a *non-linear* boundary. Empirically, we demonstrate LD-SB on three real world datasets: binary and multiclass version of Imagenette (FastAI, 2021), waterbirds-landbirds (Sagawa et al., 2020a) as well as the ImageNet (Deng et al., 2009) dataset. Compared to the results in Shah et al. (2020), our results (i) theoretically show LD-SB in a fairly general setting and (ii) empirically show LD-SB on *real* datasets.

Finally, building upon these insights, we propose a simple ensemble method – *OrthoP* – that sequentially constructs NNs by projecting out the input data directions that are used by previous NNs. We demonstrate that this method can lead to significantly more robust ensembles for real-world datasets in presence of simple distribution shifts like Gaussian noise.

**Why study $1$-hidden layer networks in the infinite width regime?**

1. From a **practical** standpoint, the dominant paradigm in machine learning right now is to pretrain large models on large amounts of data and then finetune on small target datasets. Given the large and diverse pretraining data seen by these models, it has been observed that they do learn rich features (Rosenfeld et al., 2022; Nasery et al., 2022). However, finetuning on target datasets might not utilize all the features in the pretrained model. Consequently, approaches that can train robust finetuning heads (such as a 1-hidden layer network on top) can be quite effective.
2. From a **theoretical** standpoint, there have been several works that analyze training dynamics of finite width networks (Ding et al., 2022), and show convergence to global minima on the training data. However, these results do not identify *which* among the *many* global minima, the training dynamics converge to, which is crucial in determining the nature of SB of the converged model. Such a precise characterization of the final convergence point is known only for infinite width 1-hidden layer networks (Chizat et al., 2019; Chizat & Bach, 2020).
3. While our theoretical analysis works in the setting of infinite width networks, our extensive experiments on several large scale datasets suggest that the results continue to hold even for finite width networks. Furthermore, Vyas et al. (2023) show that the behavior of neural networks remains consistent with width in the feature learning regime.

To summarize, this paper characterizes the nature of SB in 1-hidden layer networks, and also proposes a novel ensemble training approach, called OrthoP, which leads to more robust ensembles. While the theoretical results are in the infinite width regime, empirical results on several real world datasets show that these results continue to hold even for finite width networks.

**Paper organization**: This paper is organized as follows. Section 2 presents related work. Section 3 presents preliminaries. Our main results on LD-SB are presented in Section 4. Section 5 presents results on training diverse classifiers. We conclude in Section 6.

## 2 Related Work

**Simplicity Bias**: Subsequent to Shah et al. (2020), there have been several papers investigating the presence/absence of SB in various networks as well as reasons behind SB (Scimeca et al., 2021). Of these, Huh et al. (2021) is the most closely related work to ours. Huh et al. (2021) *empirically observe* that on certain *synthetic* datasets, the *embeddings* of NNs both at initialization as well as after training have a low rank structure. In contrast, we prove LD-SB *theoretically* on the IFM model as well as empirically validate this on *real* datasets. Furthermore, our results show that while the *network weights* exhibit low rank structure in the rich regime (see Section 3.2 for definition), the manifestation of LD-SB is far more subtle in lazy regime. Moreover, we also show how to use LD-SB to train a second diverse model and combine it to obtain a robust ensemble. Galanti & Poggio (2022) provide a theoretical intuition behind the relation between various hyperparameters (such as learning rate, batch size etc.) and rank of learnt weight matrices, and demonstrate it empirically. Pezeshki et al. (2021) propose that *gradient starvation* at the beginning of training is a potential reason for SB in the lazy/NTK regime but the conditions are hard to interpret. In contrast, our results are shown for any dataset in the IFM model in the *rich* regime of training. Finally, Lyu et al. (2021) consider anti-symmetric datasets and show that single hidden layer input homogeneous networks (i.e., without *bias* parameters) converge to linear classifiers. However, our results hold for general datasets and do not require input homogeneity.

**Learning diverse classifiers**: There have been several works that attempt to learn diverse classifiers. Most works try to learn such models by ensuring that the input gradients of these models do not align (Ross & Doshi-Velez, 2018; Teney et al., 2022). Xu et al. (2022) propose a way to learn diverse/orthogonal classifiers under the assumption that a complete classifier, that uses all features is available, and demonstrates its utility for various downstream tasks such as style transfer. Lee et al. (2022) learn diverse classifiers by enforcing diversity on unlabeled target data.

**Spurious correlations**: There has been a large body of work which identifies reasons for spurious correlations in NNs (Sagawa et al., 2020b) as well as proposing algorithmic fixes in different settings (Liu et al., 2021; Chen et al., 2020b). Simplicity bias seems to be one of the primary reasons behind learning spurious correlations within NNs (Shah et al., 2020).

**Implicit bias of gradient descent**: There is also a large body of work understanding the implicit bias of gradient descent dynamics. Most of these works are for standard linear (Ji & Telgarsky, 2019) or deep linear networks (Soudry et al., 2018; Gunasekar et al., 2018). For nonlinear neural networks,

one of the well-known results is for the case of $1$-hidden layer neural networks with homogeneous activation functions (Chizat & Bach, 2020), which we crucially use in our proofs. More related works are provided in Appendix.

## 3 Preliminaries

In this section, we provide the notation and background on infinite width max-margin classifiers that is required to interpret the results of this paper.

### 3.1 Basic notions

**1-hidden layer neural networks and loss function.** Consider instances $x \in \mathcal{R}^d$ and labels $y \in \{\pm 1\}$ jointly distributed as $\mathcal{D}$. A 1-hidden layer neural network model (or fully connected network (FCN)) for predicting the label for a given instance $x$, is defined by parameters $(\bar{w} \in \mathbb{R}^{m \times d}, \bar{b} \in \mathbb{R}^m, \bar{a} \in \mathbb{R}^m)$. For a fixed activation function $\phi$, given input instance $x$, the model is given as $f((\bar{w}, \bar{b}, \bar{a}), x) := \langle \bar{a}, \phi(\bar{w}x + \bar{b}) \rangle$, where $\phi(\cdot)$ is applied elementwise. The cross entropy loss $\mathcal{L}$ for a given model $f$, input $x$ and label $y$ is given as $\mathcal{L}(f(x), y) \overset{\text{def}}{=} \log(1 + \exp(-yf((\bar{w}, \bar{b}, \bar{a}), x)))$.

**Margin.** For data distribution $\mathcal{D}$, the margin of a model $f(x)$ is given as $\min_{(x,y) \sim \mathcal{D}} yf(x)$.

**Notation.** Here is some useful notation that we will use repeatedly. For a matrix $A$, $A(i, .)$ denotes the $i$th row of $A$. For any $k \in \mathbb{N}$, $\mathbb{S}^{k-1}$ denotes the surface of the unit norm Euclidean sphere in dimension $k$.

### 3.2 Initializations

The gradient descent dynamics of the network depends strongly on the scale of initialization. In this work, we primarily consider *rich regime* initialization.

**Rich regime.** In rich regime initialization, for any $i \in [m]$, the parameters $(\bar{w}(i, .), \bar{b}(i))$ of the first layer are sampled from a uniform distribution on $\mathbb{S}^d$. Each $\bar{a}(i)$ is sampled from $Unif\{-1, 1\}$, and the output of the network is scaled down by $\frac{1}{m}$ (Chizat & Bach, 2020). This is roughly equivalent in scale to Xavier initialization Glorot & Bengio (2010), where the weight parameters in both the layers are initialized approximately as $\mathcal{N}(0, \frac{2}{m})$ when $m \gg d$.

In addition, we also present some results for the lazy regime initialization described below.

**Lazy regime.** In the lazy regime, the weight parameters in the first layer are initialized with $\mathcal{N}(0, \frac{1}{d})$, those of second layer are initialized with $\mathcal{N}(0, \frac{1}{m})$ and the biases are initialized to $0$ (Bietti & Mairal, 2019; Lee et al., 2019). This is approximately equivalent in scale to Kaiming initialization (He et al., 2015).

### 3.3 Infinite Width Case

For 1-hidden layer neural networks with ReLU activation in the infinite width limit i.e., as $m \to \infty$, Jacot et al. (2018); Chizat et al. (2019); Chizat & Bach (2020) gave interesting characterizations of the trained model. As mentioned above, the training process of these models falls into one of two regimes depending on the scale of initialization (Chizat et al., 2019):

**Rich regime.** In the infinite width limit, the neural network parameters can be thought of as a distribution $\nu$ over triples $(w, b, a) \in \mathbb{S}^{d+1}$ where $w \in \mathbb{R}^d, b, a \in \mathbb{R}$. Under the rich regime initialization, the function $f$ computed by the model can be expressed as

$$f(\nu, x) = \mathbb{E}_{(w,b,a) \sim \nu}[a(\phi(\langle w, x \rangle + b)]. \tag{1}$$

Chizat & Bach (2020) showed that the training process with rich initialization can be thought of as gradient flow on the Wasserstein-2 space and gave the following characterization [3] of the trained model under the cross entropy loss $\mathbb{E}_{(x,y) \sim \mathcal{D}}[\mathcal{L}(\nu, (x, y))]$.

---

[3]Theorem 3.1 is an informal version of Chizat & Bach 2020, Theorem 5. For exact result, refer Theorem E.1 in Appendix E

**Theorem 3.1.** *(Informal)(Chizat & Bach, 2020) Under rich initialization in the infinite width limit with cross entropy loss, if gradient flow on 1-hidden layer NN with ReLU activation converges, it converges to a maximum margin classifier $\nu^*$ given as*

$$\nu^* = \arg\max_{\nu \in \mathcal{P}(\mathbb{S}^{d+1})} \min_{(x,y)\sim\mathcal{D}} y f(\nu, x), \tag{2}$$

*where $\mathcal{P}(\mathbb{S}^{d+1})$ denotes the space of distributions over $\mathbb{S}^{d+1}$.*

This training regime is known as the 'rich' regime since it learns data dependent features $\langle w, \cdot \rangle$.

**Lazy regime.** Jacot et al. (2018) showed that in the infinite width limit, the neural network behaves like a kernel machine. This kernel is popularly known as the Neural Tangent Kernel(NTK), and is given by $K(x, x') = \left\langle \frac{\partial f(x)}{\partial W}, \frac{\partial f(x')}{\partial W} \right\rangle$, where $W$ denotes the set of all trainable weight parameters. This initialization regime is called 'lazy' regime since the weights do not change much from initialization, and the NTK remains almost constant, i.e, the network does not learn data dependent features. We will use the following characterization of the NTK for 1-hidden layer neural networks.

**Theorem 3.2.** *Bietti & Mairal (2019) Under lazy regime initialization in the infinite width limit, the NTK for 1-hidden layer neural networks with ReLU activation i.e., $\phi(u) = \max(u, 0)$, is given as*

$$K(x, x') = \|x\|\|x'\|\kappa\left(\frac{\langle x, x'\rangle}{\|x\|\|x'\|}\right), where \; \kappa(u) = \frac{1}{\pi}\left(2u(\pi - cos^{-1}(u)) + \sqrt{1-u^2}\right).$$

*Lazy regime for binary classification.* Soudry et al. (2018) showed that for linearly separable datasets, gradient descent for linear predictors on logistic loss converges to the max-margin support vector machine (SVM) classifier. This implies that, any sufficiently wide neural network, when trained for a finite time in the lazy regime on a dataset that is separable by the finite-width induced NTK, will tend towards the $\mathcal{L}_2$ max-margin-classifier given by

$$\arg\min_{f \in \mathcal{H}} \|f\|_{\mathcal{H}} \text{ s.t. } yf(x) \geq 1 \; \forall \; (x, y) \sim \mathcal{D}, \tag{3}$$

where $\mathcal{H}$ represents the Reproducing Kernel Hilbert Space (RKHS) associated with the finite width kernel (Chizat, 2020). With increasing width, this kernel tends towards the infinite-width NTK (which is universal (Ji et al., 2020)). Therefore, in lazy regime, we will focus on the $\mathcal{L}_2$ max-margin-classifier induced by the infinite-width NTK.

## 4 Characterization of SB in $1$-hidden layer neural networks

In this section, we first theoretically characterize the SB exhibited by gradient descent on linearly separable datasets in the *independent features model (IFM)*. The main result, stated in Theorem 4.1, is that for binary classification of inputs in $\mathbb{R}^d$, even if there is a *single* coordinate in which the data is linearly separable, gradient descent dynamics will learn a model that relies *solely* on this coordinate, even when there are an arbitrarily large number $d-1$ of coordinates in which the data is separable, but by a non-linear classifier. In other words, the simplicity bias of these networks is characterized by *low dimensional input dependence*, which we denote by LD-SB. We then experimentally verify that NNs trained on some real datasets do indeed satisfy LD-SB.

### 4.1 Dataset

We consider datasets in the independent features model (IFM), where the joint distribution over $(x, y)$ satisfies $p(x, y) = r(y) \prod_{i=1}^{d} q_i(x_i|y)$, i.e, the features are distributed independently conditioned on the label $y$ Here $r(y)$ is a distribution over $\{-1, +1\}$ and $q_i(x_i|y)$ denotes the conditional distribution of $i^{\text{th}}$-coordinate $x_i$ given $y$. IFM is widely studied in literature, particularly in the context of naive-Bayes classifiers Lewis (1998). We make the following assumptions which posit that there are at least two features of differing complexity for classification: *one* with a linear boundary and *at least* one other with a non-linear boundary. See Figure 2 for an illustrative example.

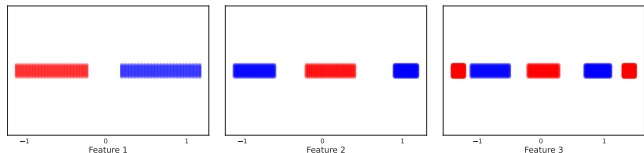

Figure 2: Illustration of an IFM dataset. Given a class $\pm 1$ represented by blue and red respectively, each coordinate value is drawn independently from the corresponding distribution. Shown above are the supports of distributions on three different coordinates for an illustrative IFM dataset, for positive and negative labels.

- One of the coordinates (say, the $1^{\text{st}}$ coordinate WLOG) is separable by a linear decision boundary [4] with margin $\gamma$ (see Figure 2), i.e, $\exists \gamma > 0$, such that $\gamma \in Supp(q_1(x_1|y = +1)) \subseteq [\gamma, \infty)$ and $-\gamma \in Supp(q_1(x_1|y = -1)) \subseteq (-\infty, -\gamma]$, where $Supp(\cdot)$ denotes the support of a distribution.
- None of the other coordinates is linearly separable. More precisely, for all the other coordinates $i \in [d] \setminus \{1\}, 0 \in Supp(q_i(x_i|y = -1))$ and $\{-1, +1\} \subseteq Supp(q_i(x_i|y = +1))$.
- The dataset can be perfectly classified even without using the linear coordinate. This means, $\exists i \neq 1$, such that $q_i(x_i|y)$ has disjoint support for $y = +1$ and $y = -1$.

Though we assume axis aligned features, our results also hold for any rotation of the dataset. While our results hold in the general IFM setting, in comparison, current results for SB e.g., Shah et al. (2020), are obtained for *very specialized* datasets within IFM[5], and do not apply to IFM in general.

## 4.2 Main result

Our main result states that, for rich initialization (Section 3.2), NNs demonstrate LD-SB for any IFM dataset satisfying the above conditions. Its proof appears in Appendix A.1.

**Theorem 4.1.** *For any dataset in the IFM model with bounded density and bounded support, satisfying the above conditions and $\gamma \geq 1$, and for $1$-hidden layer networks with ReLU activation in the infinite width limit (i.e., Eqn. (1)), there is a unique max margin classifier $\nu^*$ (i.e., satisfying Eqn. (2)). This $\nu^*$ is given by: $\nu^* = 0.5\delta_{\theta_1} + 0.5\delta_{\theta_2}$ on $\mathcal{S}^{d+1}$, where $\delta$ represents the dirac-delta distribution, $\theta_1 = (\frac{\gamma}{\sqrt{2(1+\gamma^2)}}\mathbf{e}_1, \frac{1}{\sqrt{2(1+\gamma^2)}}, 1/\sqrt{2}), \theta_2 = (-\frac{\gamma}{\sqrt{2(1+\gamma^2)}}\mathbf{e}_1, \frac{1}{\sqrt{2(1+\gamma^2)}}, -1/\sqrt{2})$ and $\mathbf{e}_1 \overset{def}{=} [1, 0, \cdots, 0]$ denotes first standard basis vector. This implies $f(\nu^*, Px^{(1)} + P_\perp x^{(2)}) = f(\nu^*, x^{(1)})$ $\forall (x^{(1)}, y^{(1)}), (x^{(2)}, y^{(2)}) \sim \mathcal{D}$, where $P$ represents the (rank-1) projection matrix on first coordinate.*

Together with Theorem 3.1, this implies that if gradient flow converges, it converges to $\nu^*$ given above. Since $P$ is a rank-1 matrix and $f(\nu^*, Px^{(1)} + P_\perp x^{(2)}) = f(\nu^*, x^{(1)})$, $\nu^*$ satisfies the first two conditions of LD-SB (Definition 1.1) with $k = 1$ and $\epsilon_1 = 0$. Moreover, since at least one of the coordinates $\{2, \ldots, d\}$ has disjoint support for $q_i(x_i|y = +1)$ and $q_i(x_i|y = -1)$, $P_\perp(x)$ can still perfectly classify the given dataset, thereby implying the third condition of LD-SB with $\epsilon_2 = 0$.

It is well known that the rich regime is more relevant for the practical performance of NNs since it allows for feature learning, while lazy regime does not (Chizat et al., 2019). Nevertheless, in the next section, we present theoretical evidence that LD-SB holds even in the lazy regime, by considering a much more specialized dataset within IFM.

## 4.3 Lazy regime

In this regime, we will work with the following dataset within the IFM family:

For $y \in \{\pm 1\}$ we generate $(x, y) \in D$ as

$$\mathbf{x}_1 = \gamma y, \; \forall i \in 2, .., d, \mathbf{x}_i = \begin{cases} \pm 1 & \text{for} \quad y = 1 \\ 0 & \text{for} \quad y = -1 \end{cases}$$

---

[4]Using linear probe for classifying pretrained representations is a standard practice in self-supervised learning (Chen et al., 2020a; Grill et al., 2020).

[5]In Shah et al. (2020), the theoretical results were obtained mainly for linear and one non-linear coordinate

Table 1: Demonstration of LD-SB in the rich regime: This table presents $P_\perp$ and $P$ logit as well as prediction changes on the five datasets. These results confirm that projection of input $x$ onto the subspace spanned by $P$ essentially determines the model's prediction on $x$. ↑ (resp. ↓) indicates that LD-SB implies a large (resp. small) value.

| Dataset | rank $(P)$ | $P_\perp$-LC $(\downarrow)$ | $P$-LC $(\uparrow)$ | $P_\perp$-pC $(\downarrow)$ | $P$-pC $(\uparrow)$ |
|---|---|---|---|---|---|
| b-Imagenette | 1 | $28.57 \pm 0.26$ | $92.13 \pm 0.24$ | $6.35 \pm 0.06$ | $47.02 \pm 0.24$ |
| Imagenette | 10 | $33.64 \pm 1.21$ | $106.29 \pm 0.53$ | $12.04 \pm 0.29$ | $89.88 \pm 0.08$ |
| Waterbirds | 3 | $25.24 \pm 1.03$ | $102.35 \pm 0.19$ | $6.78 \pm 0.15$ | $35.96 \pm 0.02$ |
| MNIST-CIFAR | 1 | $38.97 \pm 0.76$ | $101.98 \pm 0.31$ | $5.41 \pm 0.55$ | $45.15 \pm 0.44$ |
| Imagenet | 150 | $15.78 \pm 0.05$ | $132.05 \pm 0.06$ | $13.05 \pm 0.03$ | $99.76 \pm 0.01$ |

Although the dataset above is a point mass dataset, it still exhibits an important characteristic in common with the rich regime dataset – only one of the coordinates is linearly separable while others are not. For this dataset, we provide the characterization of max-margin NTK (as in Eqn. (3)):

**Theorem 4.2.** *There exists $\delta_0 > 0$ such that for every $\delta < \delta_0$, there exists an absolute constant $N$ such that for all $d > N$ and $\gamma \in [7, \delta\sqrt{d})$, the $\mathcal{L}_2$ max-margin classifier for joint training of both the layers of 1-hidden layer FCN with ReLU activation in the NTK regime on the dataset D, i.e., any f satisfying Eqn. (3) satisfies:*

$$pred(f(Px^{(1)} + P_\perp x^{(2)})) = pred(f(x^{(1)})) \qquad \forall (x^{(1)}, y^{(1)}), (x^{(2)}, y^{(2)}) \in D$$

*where $P$ represents the projection matrix on the first coordinate and $pred(f(x))$ represents the predicted label by the model $f$ on $x$.*

The proof of this theorem is presented in Appendix A.2. The above theorem shows that the prediction on a *mixed* example $Px^{(1)} + P_\perp x^{(2)}$ is the same as that on $x^{(1)}$ (i.e., $\epsilon_1 = 0$ in Definition 1.1). Furthermore, since there exists at least one coordinate $i \neq 1$ which can be used to perfectly classify the dataset, we have that Definition 1.1 is satisfied with $\epsilon_2 = 0$, thus establishing LD-SB.

## 4.4 Empirical verification

In this section, we will present empirical results demonstrating LD-SB on 4 real datasets: Imagenette (FastAI, 2021), a binary version of Imagenette (b-Imagenette), waterbirds-landbirds (Sagawa et al., 2020a) and Imagenet (Deng et al., 2009) as well as one designed dataset MNIST-CIFAR (Shah et al., 2020). More details about the datasets can be found in Appendix B.1.

### 4.4.1 Experimental setup

We take Imagenet pretrained Resnet-50 models, with 2048 features, for feature extraction and train a 1-hidden layer fully connected network, with ReLU nonlinearity. During finetuning, we freeze the backbone Resnet-50 model and train only the 1-hidden layer head (details in Appendix B.1) .

**Demonstrating LD-SB**: Given a model $f(\cdot)$, we establish its low dimensional SB by identifying a small dimensional subspace, identified by its projection matrix $P$, such that if we *mix* inputs $x_1$ and $x_2$ as $Px_1 + P_\perp x_2$, the model's output on the mixed input $\widetilde{x} \stackrel{\text{def}}{=} Px_1 + P_\perp x_2$, $f(\widetilde{x})$ is always *close* to the model's output on $x_1$ i.e., $f(x_1)$. We measure *closeness* in four metrics: (1) $P_\perp$ logit change ($P_\perp$-LC): relative change of logits wrt $x_1$ i.e., $\|f(\widetilde{x}) - f(x_1)\| / \|f(x_1)\|$, (2)$P$ logit change ($P$-LC): relative change wrt logits of $x_2$ i.e., $\|f(\widetilde{x}) - f(x_2)\| / \|f(x_2)\|$, (3) $P_\perp$-prediction change ($P_\perp$-pC): $\mathbb{P}[pred(f(\widetilde{x})) \neq pred(f(x_1))]$, and (4) $P$-prediction change ($P$-pC): $\mathbb{P}[pred(f(\widetilde{x})) \neq pred(f(x_2))]$. The quantities rank $(P)$ and $P_\perp$-pC correspond to $k$ and $\epsilon_1$ in Definition 1.1 respectively. To demonstrate that the subspace $P_\perp$ has features that are useful for prediction, we also train a new model $f_{\text{proj}}$ as follows. Given the initial model $f$ and the corresponding projection matrix $P$, we then train another model $f_{\text{proj}}$ by projecting the input through $P_\perp$ i.e., instead of using dataset $(x^{(i)}, y^{(i)})$ for training, we use $(P_\perp x^{(i)}, y^{(i)})$ for training the second model (denoted by $f_{\text{proj}}$). We refer to this training procedure as $OrthoP$ for *orthogonal projection*. The quantity $|\text{Acc}(f) - \text{Acc}(f_{\text{proj}})|$ corresponds to $\epsilon_2$ in Definition 1.1. We now describe how we identify $P$ in rich and lazy regimes.

Table 2: Demonstration of LD-SB in lazy regime: This table presents $P_\perp$ and $P$ logit as well as prediction changes on the five datasets. These results confirm that the projection of input $x$ onto the subspace spanned by $P$ essentially determines the model's prediction on $x$.

| Dataset | rank $(P)$ | $P_\perp$-LC $(\downarrow)$ | $P$-LC $(\uparrow)$ | $P_\perp$-pC $(\downarrow)$ | $P$-pC $(\uparrow)$ |
|---|---|---|---|---|---|
| b-Imagenette | 1 | $36.94\pm1.01$ | $138.41\pm1.62$ | $5.5\pm1.13$ | $47.7\pm1.55$ |
| Imagenette | 15 | $55.99\pm3.86$ | $133.86\pm5.42$ | $11.25\pm0.36$ | $89.75\pm0.15$ |
| Waterbirds | 6 | $36.89\pm5.18$ | $105.41\pm7.06$ | $20.74\pm0.64$ | $45.96\pm0.69$ |
| MNIST-CIFAR | 2 | $24.9\pm0.61$ | $141.12\pm1.86$ | $0.53\pm0.24$ | $49.83\pm0.78$ |
| Imagenet | 200 | $32.74\pm0.02$ | $132.47\pm0.04$ | $18.2\pm0.16$ | $99.74\pm0.01$ |

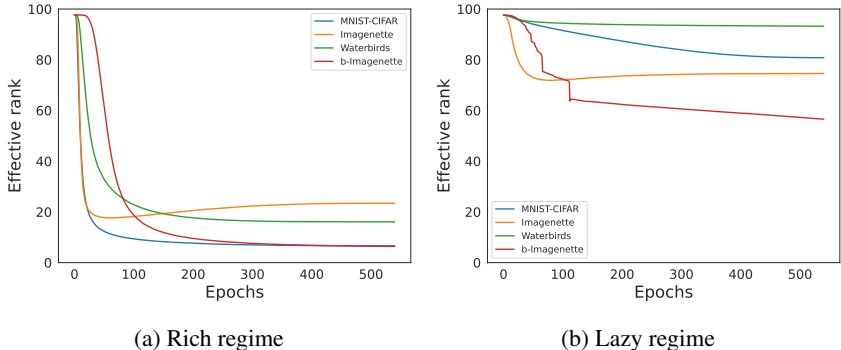

(a) Rich regime        (b) Lazy regime

Figure 3: Evolution of effective rank of first layer weight matrices in rich and lazy regimes.

### 4.4.2 Rich regime

Theorem 4.1 suggests that asymptotically, the first layer weight matrix will be low rank. However, since we train only for a finite amount of time, the weight matrix will only be approximately low rank. To quantify this, we use the notion of effective rank Roy & Vetterli (2007).

**Definition 4.3.** Given a matrix $M$, its effective rank is defined as $e^{-\sum_i \overline{\sigma_i(M)^2} \log \overline{\sigma_i(M)^2}}$ where $\sigma_i(M)$ denotes the $i^{\text{th}}$ singular value of $M$ and $\overline{\sigma_i(M)^2} \overset{\text{def}}{=} \frac{\sigma_i(M)^2}{\sum_i \sigma_i(M)^2}$.

One way to interpret the effective rank is that it is the exponential of von-Neumann entropy Petz (2001) of the matrix $\frac{MM^\top}{\text{Tr}(MM^\top)}$, where $\text{Tr}(\cdot)$ denotes the trace of a matrix. For illustration, the effective rank of a projection matrix onto $k$ dimensions equals $k$.

Figure 3a shows the evolution of the effective rank through training on the four datasets. We observe that the effective rank of the weight matrix decreases drastically towards the end of training. In this case, we set $P$ to be the subspace spanned by the top singular directions of the first layer weight matrix. Table 1 presents the results for $P_\perp$ and $P$-LC as well as pC, while Table 3 presents $\text{Acc}(f)$ and $\text{Acc}(f_{\text{proj}})$. These results establish LD-SB in the rich regime.

Table 3: Accuracy of $f_{\text{proj}}$ in rich regime

| Dataset | $\text{Acc}(f)$ | $\text{Acc}(f_{\text{proj}})$ |
|---|---|---|
| b-Imagenette | 93.35 | $91.35\pm0.32$ |
| Imagenette | 79.67 | $71.93\pm0.12$ |
| Waterbirds | 90.29 | $89.92\pm0.08$ |
| MNIST-CIFAR | 99.69 | $98.95\pm0.02$ |
| Imagenet | 72.02 | $69.63\pm0.08$ |

Table 4: Accuracy of $f_{\text{proj}}$ in lazy regime

| Dataset | $\text{Acc}(f)$ | $\text{Acc}(f_{\text{proj}})$ |
|---|---|---|
| b-Imagenette | 93.09 | $91.77\pm0.34$ |
| Imagenette | 80.31 | $77.34\pm0.21$ |
| Waterbirds | 90.4 | $89.5\pm0.18$ |
| MNIST-CIFAR | 99.74 | $98.54\pm0.00$ |
| Imagenet | 72.6 | $72.07\pm0.08$ |

Table 5: Mistake diversity and class conditioned logit correlation of models trained independently (Mist-Div $(f, f_{\text{ind}})$ and CC-LogitCorr $(f, f_{\text{ind}})$ resp.) vs trained sequentially after projecting out important features of the first model (Mist-Div $(f, f_{\text{proj}})$ and CC-LogitCorr $(f, f_{\text{proj}})$ resp.). The results demonstrate that $f$ and $f_{\text{proj}}$ are more diverse compared to $f$ and $f_{\text{ind}}$.

| Dataset | Mist-Div $(f, f_{\text{ind}})$ $(\uparrow)$ | Mist-Div $(f, f_{\text{proj}})$ $(\uparrow)$ | CC-LogitCorr $(f, f_{\text{ind}})$ $(\downarrow)$ | CC-LogitCorr $(f, f_{\text{proj}})$ $(\downarrow)$ |
|---|---|---|---|---|
| B-Imagenette | $3.87 \pm 1.54$ | $21.15 \pm 1.57$ | $99.88 \pm 0.01$ | $90.86 \pm 1.08$ |
| Imagenette | $6.6 \pm 0.46$ | $11.44 \pm 0.65$ | $99.31 \pm 0.12$ | $91 \pm 0.59$ |
| Waterbirds | $2.9 \pm 0.52$ | $14.53 \pm 0.48$ | $99.66 \pm 0.04$ | $93.81 \pm 0.48$ |
| MNIST-CIFAR | $0.0 \pm 0.0$ | $5.56 \pm 7.89$ | $99.76 \pm 0.17$ | $78.74 \pm 2.28$ |
| Imagenet | $6.97 \pm 0.06$ | $12.31 \pm 0.16$ | $99.5 \pm 0.0$ | $92.52 \pm 0.01$ |

### 4.4.3 Lazy regime

For the lazy regime, it turns out that the rank of first layer weight matrix remains high throughout training, as shown in Figure 3b. However, we are able to find a low dimensional projection matrix $P$ satisfying the conditions of LD-SB (as stated in Def 1.1) as the solution to an optimization problem. More concretely, given a pretrained model $f$ and a rank $r$, we obtain a *projection matrix* $P$ solving:

$$\min_{P} \frac{1}{n} \sum_{i=1}^{n} \left( \mathcal{L}\left( f(Px^{(i)}), y^{(i)} \right) + \lambda \mathcal{L}\left( f(P^{\perp}x^{(i)}), \mathcal{U}[L] \right) \right)$$

where $\mathcal{U}[L]$ represents a uniform distribution over all the $L$ labels, $(x^{(1)}, y^{(1)}), \cdots, (x^{(n)}, y^{(n)})$ are training examples and $\mathcal{L}(\cdot, \cdot)$ is the cross entropy loss. We reiterate that the optimization is only over $P$, while the model parameters $f$ are unchanged. In words, the above function ensures that the neural network produces correct predictions along $P$ and uninformative predictions along $P_{\perp}$. Table 2 presents the results for $P_{\perp}$ and $P$-LC as well as pC, while Table 4 presents $\text{Acc}(f)$ and $\text{Acc}(f_{\text{proj}})$. These results again establish LD-SB in the lazy regime.

## 5 Training diverse classifiers using *OrthoP*

Our results above motivate a natural strategy to construct diverse ensembles i.e., use $f$ and $f_{\text{proj}}$ instead of two independently trained models. In this section, we provide two natural diversity metrics and empirically demonstrate that $OrthoP$ leads to diverse models in practice. We also demonstrate that an ensemble of $f$ and $f_{\text{proj}}$ has higher robustness to Gaussian noise compared to an ensemble of independently trained models.

**Diversity Metrics**: Given any two models $f$ and $\tilde{f}$, we empirically evaluate their diversity using two metrics. The first is mistake diversity: Mist-Div $\left( f, \tilde{f} \right) \stackrel{\text{def}}{=} 1 - \frac{|\{i: f(\mathbf{x}^{(i)}) \neq y^{(i)} \ \& \ \tilde{f}(\mathbf{x}^{(i)}) \neq y^{(i)}\}|}{\min(|\{i: f(\mathbf{x}^{(i)}) \neq y^{(i)}\}|, |\{i: \tilde{f}(\mathbf{x}^{(i)}) \neq y^{(i)}\}|)}$, where we abuse notation by using $f(x_i)$ (resp. $\tilde{f}(x_i)$) to denote the class predicted by $f$ (resp $\tilde{f}$) on $x_i$. Higher Mist-Div $\left( f, \tilde{f} \right)$ means that there is very little overlap in the mistakes of $f$ and $\tilde{f}$. The second is class conditioned logit correlation i.e., correlation between outputs of $f$ and $\tilde{f}$, conditioned on the class. More concretely, CC-LogitCorr $\left( f, \tilde{f} \right) = \frac{\sum_{y \in \mathcal{Y}} \text{Corr}([f(\mathbf{x}_i)], [\tilde{f}(\mathbf{x}_i)]: y_i = y)}{|\mathcal{Y}|}$, where corr$([f(\mathbf{x}_i)], [\tilde{f}(\mathbf{x}_i)] : y_i = y)$ represents the empirical correlation between the logits of $f$ and $\tilde{f}$ on the data points where the true label is $y$. Table 5 compares the diversity of two independently trained models ($f$ and $f_{\text{ind}}$) with that of two sequentially trained models ($f$ and $f_{\text{proj}}$). The results demonstrate that $f$ and $f_{\text{proj}}$ are more diverse compared to $f$ and $f_{\text{ind}}$. We have also compared $OrthoP$ to another diverse training method Evading-SB (Teney et al., 2022). The results are provided in Table 7 in the Appendix. As can be seen, our results are either better or comparable to Evading-SB.

**Ensembling**: Figure 4 shows the variation of test accuracy with the strength of gaussian noise added to the pretrained representations of the dataset. Here, an ensemble is obtained by weighted averaging of the logits of multiple models, trained either independently ($f_{\text{ind}}$) or using $OrthoP$ ($f_{\text{proj}}$). Moreover, we also compare our method to another diversity training method ($f_{\text{esb}}$) termed as

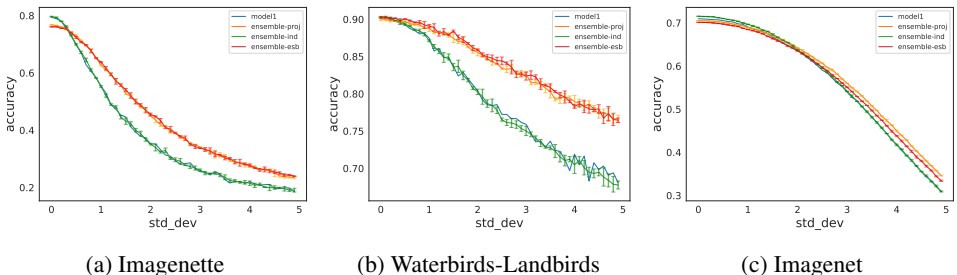

|  |  |  |
|---|---|---|
| (a) Imagenette | (b) Waterbirds-Landbirds | (c) Imagenet |

Figure 4: Variation of test accuracy vs standard deviation of Gaussian noise added to the pretrained representations of the dataset. Here *model1*, *ensemble-ind*, *ensemble-proj* and *ensemble-esb* refer to the original model $f$, ensemble of $f$ and independently trained model $f_{\text{ind}}$, ensemble of $f$ and $f_{\text{proj}}$ trained using OrthoP, and ensemble of $f$ and $f_{\text{esb}}$ obtained using Evading-SB method respectively.

Evading-SB (Teney et al., 2022). We can see that, an ensemble of $f$ and $f_{\text{proj}}$ is much more robust as compared to an ensemble of $f$ and $f_{\text{ind}}$, and generally comparable to and ensemble of $f$ and $f_{\text{esb}}$.

## 6 Conclusion: Summary, Limitations and Future Directions

In this work, we propose a rigorous definition of simplicity bias, which is believed to be a key reason for their brittleness (Shah et al., 2020). In particular, we prove that 1-hidden layer networks suffer from low dimensional input dependence (LD-SB), and empirically verify this phenomenon on several real world datasets. We also propose a novel approach – OrthoP– to train diverse models, and demonstrate that an ensemble consisting of such diverse models is more robust to Gaussian noise. Extending these inisights to deeper models or in the finite width setting are interesting directions for future work.

## 7 Acknowledgements

We acknowledge support from Simons Investigator Fellowship, NSF grant DMS-2134157, DARPA grant W911NF2010021, and DOE grant DE-SC0022199.

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

# A   Proofs for rich and lazy regime

## A.1   Rich regime

We restate Theorem 4.1 below and prove it.

**Theorem A.1.** *For any dataset in IFM model satisfying the conditions in Section 4.1, $\gamma \geq 1$ and $f(\nu, x)$ as in Eqn. (1), the distribution $\nu^* = 0.5\delta_{\theta_1} + 0.5\delta_{\theta_2}$ on $\mathcal{S}^{d+1}$ is the unique max-margin classifier satisfying Eqn. (2), where $\theta_1 = (\frac{\gamma}{\sqrt{2(1+\gamma^2)}}\mathbf{e}_1, \frac{1}{\sqrt{2(1+\gamma^2)}}, 1/\sqrt{2}), \theta_2 = (-\frac{\gamma}{\sqrt{2(1+\gamma^2)}}\mathbf{e}_1, \frac{1}{\sqrt{2(1+\gamma^2)}}, -1/\sqrt{2})$ and $\mathbf{e}_1 \overset{def}{=} [1, 0, \cdots, 0]$ denotes first standard basis vector. In particular, this implies that if gradient flow for 1-hidden layer FCN with ReLU activation under rich initialization in the infinite width limit with cross entropy loss converges, and satisfies the technical conditions in Theorem E.1, then it converges to $\nu^*$ satisfying $f(\nu^*, Px_1 + P_\perp x_2) = f(\nu^*, x_1)\forall(x_1, y_1), (x_2, y_2) \in D$, where $P$ represents the (rank-1) projection matrix on the first coordinate.*

*Proof of Theorem A.1:* The proof relies on showing that $\nu^*$ is a max-margin classifier as in Theorem 3.1. To this end, we employ a primal-dual characterization of max-margin classifiers and construct a dual certificate that proves the optimality of margin of $\nu^*$. Chizat & Bach (2020) showed the following primal-dual characterization of maximum margin classifiers in eqn. (2):

**Lemma A.2.** *(Chizat & Bach, 2020) $\nu^*$ satisfies eqn. (2) if there exists a data distribution $p^*$ such that the following two complementary slackness conditions hold:*

$$Supp(\nu^*) \subseteq \underset{(w,b,a)\in\mathbb{S}^{d+1}}{\arg\max} \; \mathbb{E}_{(x,y)\sim p^*} y[a(\phi(\langle w, x\rangle + b))] \quad and \tag{4}$$

$$Supp(p^*) \subseteq \underset{(x,y)\sim\mathcal{D}}{\arg\min} \; y\mathbb{E}_{(w,b,a)\sim\nu^*}[a(\phi(\langle w, x\rangle + b))] \,. \tag{5}$$

The plan is to construct a distribution $p^*$ that satisfies the conditions of the above Lemma.

**Uniqueness.** Note further that for a fixed $p^*$, $\mathbb{E}_{(x,y)\sim p^*} yf(\nu, x)$ is an upper bound for the margin $\min_{(x,y)\sim\mathcal{D}} yf(\nu, x)$ of any classifier $\nu$. Hence, for uniqueness, it suffices to show that $\delta_{\theta_1}, \delta_{\theta_2}$ are the unique maximizers of the objective on the RHS of eqn. (4) and that the unique maximum margin convex combination of $\delta_{\theta_1}, \delta_{\theta_2}$ over $\mathcal{D}$ is $\nu^*$.

We first describe the support $D$ of $p^*$. For $y \in \{\pm 1\}$ we generate $(x, y) \in D$ as

$$\mathbf{x}_1 = \gamma y$$

$$\forall i \in 2, .., d, \mathbf{x}_i = \begin{cases} \pm 1 & \text{for} \quad y = 1 \\ 0 & \text{for} \quad y = -1 \end{cases}$$

Now for $(x, y) \in D$, define

$$p^*(x, y) = \begin{cases} 0.5 & \text{for} \quad y = 1 \\ 0.5^d & \text{for} \quad y = -1 \end{cases} \tag{6}$$

Note that $p^*$ is supported on $2^{d-1}$ positive instances and one negative instance. We begin by showing eqn. (5).

*Claim* A.3. $p^*$ as in eqn. (6) satisfies eqn. (5). Further, the unique maximum margin convex combination of $\delta_{\theta_1}, \delta_{\theta_2}$ is $\nu^*$.

*Proof.* Let us find the minimizers $(x, y) \sim \mathcal{D}$ of $yf(\nu, x) = y\mathbb{E}_{(w,b,a)\sim\nu^*}[a(\phi(\langle w, x\rangle + b))]$ for any $\nu = \lambda\delta_{\theta_1} + (1 - \lambda)\delta_{\theta_2}, 0 \leq \lambda \leq 1$.

$yf(\nu, x)$ for $(x, y)$ with $y = -1$ (denoting $x_1$ by $-\alpha_1$, where $\alpha_1 \geq \gamma$) is

$$yf(\nu, x) = -1\Big[\lambda * \phi\left(\frac{\gamma}{\sqrt{2(1+\gamma^2)}}\mathbf{e}_1^\top(-\alpha_1\mathbf{e}_1) + \frac{1}{\sqrt{2(1+\gamma^2)}}\right) * \frac{1}{\sqrt{2}}$$

$$+ (1-\lambda) * \phi\left(-\frac{\gamma}{\sqrt{2(1+\gamma^2)}}\mathbf{e}_1^\top(-\alpha_1\mathbf{e}_1) + \frac{1}{\sqrt{2(1+\gamma^2)}}\right) * \frac{-1}{\sqrt{2}}\Big],$$

and for $(x, y)$ with $y = 1$ (denoting $x_1$ by $\alpha_2$, where $\alpha_2 \geq \gamma$) is

$$yf(\nu, x) = 1\left[\lambda * \phi\left(\frac{\gamma}{\sqrt{2(1+\gamma^2)}}\mathbf{e}_1^\top(\alpha_2\mathbf{e}_1) + \frac{1}{\sqrt{2(1+\gamma^2)}}\right) * \frac{1}{\sqrt{2}}\right.$$

$$\left. + (1-\lambda) * \phi\left(-\frac{\gamma}{\sqrt{2(1+\gamma^2)}}\mathbf{e}_1^\top(\alpha_2\mathbf{e}_1) + \frac{1}{\sqrt{2(1+\gamma^2)}}\right) * \frac{-1}{\sqrt{2}}\right].$$

As $\gamma \geq 1$, the expressions above equal $\lambda\frac{\sqrt{\gamma\alpha_1+1}}{2}$ and $(1-\lambda)\frac{\sqrt{\gamma\alpha_2+1}}{2}$ respectively, and hence are minimized at $\alpha_1 = \alpha_2 = \gamma$. Hence, the margin of $\nu$ is $\min(\lambda, 1-\lambda)\frac{\sqrt{1+\gamma^2}}{2}$ which is uniquely maximized at $\lambda = 1/2$. Further for $\lambda = 1/2$, all points in $D$ have the same value of $yf(\nu, x)$. $\square$

In the rest of the proof we show eqn. (4), Let us denote by $g(w, b, a) \coloneqq \mathbb{E}_{(x,y)\sim p^*}y[a(\phi(\langle w, x\rangle + b))]$. We show that $\delta_{\theta_1}, \delta_{\theta_2}$ are the only maximizers of $g(w, b, a)$ over $\mathbb{S}^{d+1}$.

We first find $g(\theta_1), g(\theta_2)$.

$$g(\theta_1) = \Pr(y = 1) \cdot 1 \cdot \frac{1}{\sqrt{2}} \cdot \phi\left(\frac{\gamma}{\sqrt{2(1+\gamma^2)}}\mathbf{e}_1^T(\gamma\mathbf{e}_1) + \frac{1}{\sqrt{2(1+\gamma^2)}}\right)$$

$$+ \Pr(y = -1) \cdot -1 \cdot \frac{1}{\sqrt{2}} \cdot \phi\left(\frac{\gamma}{\sqrt{2(1+\gamma^2)}}\mathbf{e}_1^T(-\gamma\mathbf{e}_1) + \frac{1}{\sqrt{2(1+\gamma^2)}}\right) = \frac{\sqrt{\gamma^2+1}}{4},$$

where the first term is because $w_2, w_3, \ldots, w_d$ are zero for $\theta_1$. Similarly, $g(\theta_2) = \frac{\sqrt{\gamma^2+1}}{4}$. We now show that $g(w, a, b) < \frac{\sqrt{\gamma^2+1}}{4}$ for $(w, a, b) \notin \{\theta_1, \theta_2\}$.

We begin by showing the following simple but useful claim.

*Claim* A.4. All maximizers of $g(w, b, a)$ over $\mathbb{S}^{d+1}$ satisfy $|a| = 1/\sqrt{2}$.

*Proof.* The proof essentially follows from the $1-$homogeneity of the ReLU function $\phi$ and separability of $g(w, b, a)$. Note that $g(w, b, a) = \sqrt{\|w\|^2 + b^2}a \cdot g(w', b', 1)$ where $\|w'\|^2 + b^2 = 1$. Maximizing $g(w, b, a)$ is equivalent to maximizing $g(w', b', 1)$ over $\mathbb{S}^d$ and $a\sqrt{\|w\|^2 + b^2}$ over $\mathbb{S}^{d+1}$ respectively. The second of these has its unique maximum at $|a| = 1/\sqrt{2}$, completing the proof. $\square$

Now express $g(w, b, a)$ as

$$g(w, b, a) = a\left(\Pr(y = 1)\mathbb{E}[\phi(w^Tx + b)|y = 1] - \Pr(y = -1)\mathbb{E}[\phi(w^Tx + b)|y = -1]\right)$$

$$= \frac{a}{2}\left(\mathbb{E}_\sigma[\phi(\gamma w_1 + b + \sum_{i=2}^{d}\sigma_i w_i)] - \phi(b - \gamma w_1)\right), \tag{7}$$

where $\sigma_i$ are independent Rademacher random variables. We have two cases on $a$:

**Case 1:** $a = 1/\sqrt{2}$. By eqn. (7) we have

$$g(w, b, 1/\sqrt{2}) \leq \frac{1}{2\sqrt{2}}\mathbb{E}_\sigma\left[\phi(\gamma w_1 + b + \sum_{i=2}^{d}\sigma_i w_i)\right].$$

To simplify the above, define the random variable $X = \sum_{i=2}^{d}\sigma_i w_i$ and denote $\gamma w_1 + b$ by $\alpha$. Note that $|\alpha| = |\gamma w_1 + b| \leq \sqrt{\frac{\gamma^2+1}{2}}$ which follows from $\|w\|^2 + b^2 = 1/2$. The expectation in the last expression above becomes

$$\mathbb{E}[\phi(X + \alpha)] = \mathbb{E}[(X + \alpha)\mathbf{1}\{X + \alpha \geq 0\}] = \mathbb{E}[X\mathbf{1}\{X \geq -\alpha\}] + \alpha\Pr(X \geq -\alpha)$$
$$= \mathbb{E}[X\mathbf{1}\{X \geq \alpha\}] + \alpha(1 - \Pr(X \geq \alpha)) \leq \mathbb{E}[X\mathbf{1}\{X \geq \alpha\}] + \alpha,$$

where the last equality follows from symmetry of $X$. Note that $Var(X) = \sum_{i=2}^{d} w_i^2$ which is at most $\frac{1}{2} - \frac{\alpha^2}{1+\gamma^2}$ (using $\gamma w_1 + b = \alpha$ and $\|w\|^2 + b^2 = 1/2$). Using A.5 to upper bound $\mathbb{E}[X\mathbf{1}\{X \geq \alpha\}]$ we have

$$\mathbb{E}[\phi(X + \alpha)] \leq \alpha + \sqrt{\frac{1}{2} \min\left(\frac{1}{2}, \frac{\frac{1}{2} - \frac{\alpha^2}{1+\gamma^2}}{2\alpha^2}\right)\left(\frac{1}{2} - \frac{\alpha^2}{1+\gamma^2}\right)}.$$

We can check that the RHS of the above has its unique maximizer at $\alpha = \sqrt{\frac{1+\gamma^2}{2}}$ for $|\alpha| \leq \sqrt{\frac{1+\gamma^2}{2}}$. Hence $g(w, b, a) \leq \frac{\sqrt{1+\gamma^2}}{4}$ in this case. We are now done since any $(w_1, b)$ satisfying $\gamma w_1 + b = \sqrt{\frac{1+\gamma^2}{2}}$ and $w_1^2 + b^2 \leq 1/2$ has $b = \frac{1}{\sqrt{2(1+\gamma^2)}}$.

**Case 2:** $a = -1/\sqrt{2}$. Using eqn. (7) we have $g(w, b, -1/\sqrt{2}) \leq \phi(b - \gamma w_1)/2\sqrt{2}$ which for $b^2 + w_1^2 \leq 1/2$ attains its unique maximum $\sqrt{\frac{\gamma^2+1}{4}}$ at $b = \frac{1}{\sqrt{2(1+\gamma^2)}}$.

Finally, note that the weights of the *trained* network $(w, b, a)$ are sampled from $\nu^*$. Hence, the final claim in the theorem about $f(\nu^*, Px_1 + P_\perp x_2)$ follows since the distribution of $w$ only has a support on $\mathbf{e}_1$ and $-\mathbf{e}_1$.

$\square$

**Note:** The proof above also holds for any rotation of the dataset, with the weights corresponding to the first standard basis vector, i.e, the direction of linearly separable coordinate in that basis.

### A.1.1 Auxiliary lemmas for rich regime

**Lemma A.5.** *For any symmetric discrete random variable X with bounded variance, for $\alpha > 0$,*

$$\mathbb{E}[X\mathbb{I}(X \geq \alpha)] \leq \sqrt{\frac{1}{2} \min\left(\frac{1}{2}, \frac{Var(X)}{2\alpha^2}\right) Var(X)}.$$

*Proof.*

$$\mathbb{E}[X\mathbb{I}(X \geq \alpha)] = \sum_{x \geq \alpha} xp(x) = \sum_{x \geq \alpha} \sqrt{p(x)}\sqrt{p(x)}x \leq \sqrt{p(X \geq \alpha)\sum_{x \geq \alpha} x^2 p(x)}, \quad (8)$$

where the last inequality is by Cauchy-Schwartz. Also by Chebyshev's inequality, $p(|X| \geq \alpha) \leq Var(X)/2\alpha^2$. Combining this with eqn. (8) and using symmetry of $X$ and non-negativity of $\alpha$ gives the required lemma. $\square$

### A.1.2 $OrthoP$ method on IFM

Here, we theoretically establish that $f$ and $f_{\text{proj}}$ obtained via $OrthoP$ rely on different features for any dataset within IFM. Consequently, by the definition of IFM, $f$ and $f_{\text{proj}}$ have independent logits conditioned on the class.

**Proposition A.6.** *Consider any IFM dataset as described in Section 4.1. Let $f$ be the model described in Theorem 3.1 and $f_{proj}$ be the second model obtained by $OrthoP$. Then, the outputs $f$ and $f_{proj}$ on $x$ i.e., $f(x)$ and $f_{proj}(x)$ depend only on $x_1$ and $\{x_2, \cdots, x_d\}$ respectively. Let the model obtained in Theorem 3.1 be denoted by $f$. Consider the projection matrix $P$ along the top singular vector of the first layer weight matrix of $f$. Then, the dataset obtained by projecting the input through $P_\perp$ is not separable along the linear coordinate.*

*Proof.* As shown in Theorem 3.1, the final distribution of the weights is given by $\nu^* = 0.5\delta_{\theta_1} + 0.5\delta_{\theta_2}$, where $\theta_1 = (\frac{\gamma}{\sqrt{2(1+\gamma^2)}}\mathbf{e}_1, \frac{1}{\sqrt{2(1+\gamma^2)}}, 1/\sqrt{2})$, $\theta_2 = (-\frac{\gamma}{\sqrt{2(1+\gamma^2)}}\mathbf{e}_1, \frac{1}{\sqrt{2(1+\gamma^2)}}, -1/\sqrt{2})$ and $\mathbf{e}_1 \stackrel{\text{def}}{=} [1, 0, \cdots, 0]$ denotes first standard basis vector.

As the first layer weight matrix only has support along the $\mathbf{e}_1$ direction, therefore its top singular vector also points along the $\mathbf{e}_1$ direction. Hence, $P = \mathbf{e}_1 \mathbf{e}_1^\top$ and $P_\perp = I - \mathbf{e}_1 \mathbf{e}_1^\top$, where $I$ denotes the identity matrix. Thus, the dataset obtained by projecting the input through $P_\perp$ has value 0 for the linear coordinate, for both $y = +1$ and $y = -1$. Hence, it is not separable along the linear coordinate. Thus, the second model $f_{proj}$ relies on other coordinates for classification. $\qquad\square$

### A.2 Lazy regime

Theorem 4.2 is a corollary of the following more general theorem.

**Theorem A.7.** *Consider a point $x \in D$. For sufficiently small $\epsilon > 0$, there exist an absolute constant $N$ such that for all $d > N, \gamma < \epsilon\sqrt{d}$ and $\gamma \geq 7$, for the joint training of both the layers of 1-hidden layer FCN with ReLU activation in the NTK regime, the prediction of any point of the form $(\zeta, x_{2:d})$ satisfies the following:*

*1. For $\zeta \geq 0.73$, the prediction is positive.*

*2. For $\zeta \leq -0.95\gamma$, the prediction is negative.*

The above theorem establishes that perturbing $x_1$ by $O(\gamma)$ changes $pred(f(x))$ for $x \in D$ (whereas a classifier exists that achieves a margin of $\Omega(\sqrt{d})$ on $D$, as $D$ has margin 1 for coordinates $\{2 \cdots d\}$). As $\gamma = o(d)$, this shows that the learned model is adversarially vulnerable.

*Proof of Theorem A.7.* The idea of the proof is to obtain an analytical expression for $f(x)$ using KKT conditions for the max-margin SVM for the NTK kernel (as in Theorem 3.2).

We begin with some preliminaries. We will refer to the first coordinate of the instance as the 'linear' coordinate, and to the rest as 'non-linear' coordinates. Also, henceforth we append an extra coordinate with value 1 to all our instances (corresponding to bias term) - as is standard for working with unbiased SVM without loss of generality.

**Explicit expression for $f$.** Using representer theorem for max margin kernel SVM, we know that $f$ can be expressed as

$$f(x) = \sum_{(x^{(t)}, y^{(t)}) \in D} \lambda_t y^{(t)} K(x, x^{(t)}),$$

for some $\lambda_t \geq 0$ (that are known as *Lagrange multipliers*). Further by KKT conditions, a function possessing such a representation (that correctly classifies $D$) has maximum margin if $y^{(t)} f(x^{(t)}) = 1$ whenever $\lambda_t > 0$ (training points $t$ satisfying $\lambda_t > 0$ are called *support vectors*).

We begin with a useful claim.

*Claim A.8.* The max margin kernel SVM for $D$ with the NTK kernel has all points in $D$ as support vectors.

*Proof.* By the above discussion, it suffices to show that the (unique) solution $\alpha \in \mathbb{R}^{|D|}$ to $K\alpha = y$ satisfies $\text{sign}(\alpha_i) = y^{(i)}$ for all $i$, where $K$ is the $|D| \times |D|$ Gram matrix with $(i,j)$th entry $K(x^{(i)}, x^{(j)})$ and $y_i = y^{(i)}$ (the Lagrange multipliers $\lambda_i$ are then given by $y_i \alpha_i$).

*Structure of Gram matrix.* Order $D$ so that the positive instances appear first. Then the Gram matrix $K$ has a block structure of the form $\begin{pmatrix} B & C \\ C^T & R \end{pmatrix}$ where $B \in \mathbb{R}^{2^{d-1} \times 2^{d-1}}$ and $R \in \mathbb{R}$ are the Gram matrices for the positive and negative instances respectively, and $C \in \mathbb{R}^{2^{d-1} \times 1}$ represents the $K(x^{(i)}, x^{(|D|)})$ values for $i < |D|$.

Recall that for the NTK kernel, $K(x^{(i)}, x^{(j)})$ has the form $\|x^{(i)}\|\|x^{(j)}\|\kappa(\langle x^{(i)}, x^{(j)} \rangle)$. Note all the positive instances have the same norm (denoted by $\rho_1 = \sqrt{d + \gamma^2}$) and the inner product between two positive instances depends only on the number $i$ of non-matching non-linear coordinates (denoted by $\beta_i$ for $0 \leq i \leq d-1$). Hence, the rows of $B$ are permutations of each other, with the entry $\rho_1^2 \beta_i$ appearing $\binom{d-1}{i}$ times. Similarly, the entries in $C$ are all equal and are denoted by $\rho_1 \rho_2 \beta_d$ where $\beta_d$

denotes $\kappa(x^{(t)}, x^{|D|})$ for any $t < |D|$ and $\rho_2 = \|x^{|D|}\| = \sqrt{1+\gamma^2}$. The only entry in $R$ is $\rho_2^2\kappa(1)$. In particular,

$$\beta_i = \kappa\left(\frac{d - 2i + \gamma^2}{d + \gamma^2}\right) \text{ for } i \in [|D| - 1], \quad \text{and} \quad \beta_d = \kappa\left(\frac{1 - \gamma^2}{\sqrt{d + \gamma^2}\sqrt{1 + \gamma^2}}\right).$$

Now we are ready to solve $K\alpha = y$. By symmetry in the structure of K, $\alpha$ looks like $[a, a, ......, b]$, where the first $|D| - 1$ entries are the same.

Expanding $K\alpha = y$, we get two equations given by

$$a\rho_1^2\left(\sum_{i=0}^{d-1}\binom{d-1}{i}\beta_i\right) + b\rho_1\rho_2\beta_d = 1 \quad \text{and} \quad 2^{d-1}a\rho_1\rho_2\beta_d + \rho_2^2\kappa(1)b = -1.$$

Solving, we get

$$a = \frac{\rho_2\kappa(1) + \rho_1\beta_d}{\rho_1^2\rho_2\sum_{i=0}^{d-1}\left(\binom{d-1}{i}[\kappa(1)\beta_i - \beta_d^2]\right)} \quad \text{and} \quad b = \frac{-1 - 2^{d-1}a\rho_1\rho_2\beta_d}{\rho_2^2\kappa(1)}.$$

We now show that $a > 0$ and $b < 0$. Note that for sufficiently large $d$, $\beta_d$ can be made arbitrarily close to $\kappa(0) = 1/\pi$ (since $\kappa$ is smooth around 0). Hence, $a > 0$ implies $b < 0$. We in fact give the following estimate for $a$:

$$a = 2^{1-d} \cdot \frac{\rho_2\kappa(1) + \rho_1\beta_d}{\xi\rho_1^2\rho_2} \quad \text{where} \quad \frac{2}{\pi} - \frac{1}{\pi^2} + O\left(\frac{1}{d}\right) \le \xi \le 2 + O\left(\frac{1}{d}\right). \quad (9)$$

For the lower bound on $\xi$, write

$$\sum_{i=0}^{d-1}\binom{d-1}{i}[\kappa(1)\beta_i - \beta_d^2] = \kappa(1)\sum_{i=0}^{\lfloor d/2 \rfloor}\binom{d-1}{i}(\beta_i + \beta_{d-1}) - 2^{d-1}\beta_d^2$$

$$\ge \kappa(1)\sum_{i=0}^{\lfloor d/2 \rfloor}\binom{d-1}{i}2\beta_{d/2} - 2^{d-1}\beta_d^2 \ge 2^{d-1}\left(\kappa(1)\kappa(0) - \kappa^2(0) + O\left(\frac{1}{d}\right)\right),$$

where for the first inequality we used convexity of $\kappa$ and for the second inequality we used $\beta_{d/2} = \kappa(0) + O(1/d), \beta_d = \kappa(0) + O(1/\sqrt{d})$. For the upper bound on $\xi$, write

$$\sum_{i=0}^{d-1}\binom{d-1}{i}[\kappa(1)\beta_i - \beta_d^2] \le \kappa(1)\sum_{i=0}^{d-1}\binom{d-1}{i}\kappa\left(1 - \frac{2i}{d + \gamma^2}\right)$$

$$\le \kappa(1)\sum_{i=0}^{d-1}\binom{d-1}{i}\left(2 - \frac{2i}{d + \gamma^2}\right) = \kappa(1)2^d - \frac{\kappa(1)(d-1)2^{d-1}}{d + \gamma^2},$$

where for the second inequality we used $\kappa(u) \le 1 + u$ (which holds by convexity and $\kappa(-1) = 0, \kappa(1) = 2$). $\qquad\square$

Now we analyze predicted labels for points of the form $(\zeta, x_{2:d+1})$ where $x \in D$. We make two cases depending on the label of $x$.

**Predicted label for point $(\zeta, x_{2:d+1}^{(t)})$ where $x^{(t)} \in D$ has positive label**

Our point (denoted by $x$) has the form $(\zeta, \zeta_1, \zeta_2, \ldots, \zeta_d, 1)$ where $\zeta_i \in \pm 1$. The idea of the proof is to write $f$ explicitly as a function of $\zeta$ and work with its first order Taylor expansion around $\zeta = \gamma$, with some additional work to take care of non-smoothness of $f$.

*Explicit form for $f$.* Let $\tau_i \overset{\text{def}}{=} \langle x, x'\rangle/(\|x\|\|x'\|)$ for a positive instance $x' \in D$, where $x$ and $x'$ have exactly $i$ non-matching non-linear coordinates (for $0 \le i \le d - 1$). Similarly denote by $\tau_d$ the quantity $\langle x, x^{|D|}\rangle/(\|x\|\|x^{|D|}\|)$. In particular,

$$\tau_i = \left(\frac{d - 2i + \gamma\zeta}{\rho_1\|x\|}\right) \quad \text{and} \quad \tau_d = \left(\frac{1 - \gamma\zeta}{\rho_2\|x\|}\right).$$

By the above discussion, we have

$$f(x) = a \left( \sum_{t=1}^{|D|-1} K(x, x^{(t)}) \right) + bK(x, x^{|D|}) = a\rho_1 \|x\| \left( \sum_{i=0}^{d-1} \binom{d-1}{i} \kappa(\tau_i) \right) + b\rho_2 \|x\| \kappa(\tau_d).$$

Substituting $b$ and denoting $f(x)/\|x\|$ by $g(\zeta)$ we get

$$g(\zeta) = a\rho_1 \left[ \sum_{i=0}^{d-1} \binom{d-1}{i} \kappa(\tau_i(\zeta)) - \frac{2^{d-1}\beta_d}{\kappa(1)} \kappa(\tau_d(\zeta)) \right] - \frac{\kappa(\tau_d(\zeta))}{\rho_2 \kappa(1)}. \tag{10}$$

Now try to expand $g(\zeta)$ using the Taylor series around $\zeta = \gamma$ (note that $g(\gamma) = 1/\rho_1$). Note that $\kappa'$ can however be unbounded around $-1$ and $1$. To get around this, write $g = h + q$, where $h$ has bounded first and second derivative, and $q$ has lower order than $h$ for $\zeta$ of interest. In particular,

$$h(\zeta) = a\rho_1 \left[ \sum_{i=d/4}^{3d/4} \binom{d-1}{i} \kappa(\tau_i(\zeta)) - \frac{2^{d-1}\beta_d}{\kappa(1)} \kappa(\tau_d(\zeta)) \right] - \frac{\kappa(\tau_d(\zeta))}{\rho_2 \kappa(1)} \qquad \text{and}$$

$$q(\zeta) = a\rho_1 \left[ \sum_{i:|d/2-i|>d/4} \binom{d-1}{i} \kappa(\tau_i(\zeta)) \right].$$

Observe that $q(\zeta) = o(c^d)$ for $c < 1$ using the estimate eqn. (9) for $a$ and concentration for sums of independent Bernoullis. By Taylor's theorem,

$$g(\zeta) = h(\gamma) + h'(\gamma)(\zeta - \gamma) + \frac{h''(\theta)(\zeta - \gamma)^2}{2} + q(\zeta), \tag{11}$$

for some $\theta \in [\gamma, \zeta]$, where $h(\gamma) \approx 1/\sqrt{d}$. It will turn out that $|h'(\gamma)| = \Theta(1/\sqrt{d})$, $|h''(\zeta)| = o(1/\sqrt{d})$. This will allow us to complete the proof using the linear approximation of $g(\zeta)$ by neglecting the second order term and $q(\zeta)$. We now compute $h', h''$, treating $\|x\| = \sqrt{d + \zeta^2}$ as a constant for exposition (the proof works without this approximation or the reader may think of $\gamma$ as $o(\sqrt{d})$). Using $\tau_i'(\zeta) \approx \frac{\gamma}{\rho_1 \|x\|}, \tau_d'(\zeta) \approx \frac{-\gamma}{\rho_2 \|x\|}$,

$$h'(\zeta) \approx a\rho_1 \left[ \sum_{i=0}^{d-1} \binom{d-1}{i} \kappa'(\tau_i(\zeta)) \frac{\gamma}{\rho_1 \|x\|} + \frac{2^{d-1}\beta_d}{\kappa(1)} \kappa'(\tau_d(\zeta)) \frac{\gamma}{\rho_2 \|x\|} \right] + \frac{\kappa'(\tau_d(\zeta))}{\rho_2 \kappa(1)} \frac{\gamma}{\rho_2 \|x\|}$$

$$h''(\zeta) \approx a\rho_1 \left[ \sum_{i=0}^{d-1} \binom{d-1}{i} \kappa''(\tau_i(\zeta)) \frac{\gamma^2}{\rho_1^2 \|x\|^2} - \frac{2^{d-1}\beta_d}{\kappa(1)} \kappa''(\tau_d(\zeta)) \frac{\gamma^2}{\rho_2^2 \|x\|^2} \right] - \frac{\kappa''(\tau_d(\zeta))}{\rho_2 \kappa(1)} \frac{\gamma^2}{\rho_2^2 \|x\|^2}.$$

Plugging $\|x\| \approx \rho_1 \approx \sqrt{d}$ and substituting $a$ from eqn. (9),

$$h'(\zeta) = \frac{(1 + \beta_d^2/\xi)\kappa'(\tau_d(\zeta))\gamma}{\rho_2^2 \kappa(1)\sqrt{d}} + o\left( \frac{1}{\sqrt{d}} \right) \qquad \text{and} \qquad h''(\zeta) = O\left( \frac{1}{d} \right),$$

which substituted in eqn. (11) with $\tau_d(\zeta) \approx 0, \beta_d \approx \kappa(0), \kappa'(\tau_d(\zeta)) \approx \kappa'(0)$ gives

$$g(\zeta) = \frac{1}{\sqrt{d}} \left( 1 + \frac{(1 + \kappa^2(0)/\xi)\kappa'(0)\gamma}{\kappa(1)\rho_2^2}(\zeta - \gamma) \right) + o\left( \frac{1}{\sqrt{d}} \right),$$

Hence, $g(\zeta) > 0$ whenever the coefficient of $1/\sqrt{d}$ above is bounded above zero, and a similar condition holds for $g(\zeta) < 0$. Using the estimates of $\xi$ from eqn. (9) and $\kappa'(0) = 1, \kappa(0) = 1/\pi, \kappa(1) = 2, \rho_2^2 = 1 + \gamma^2$ in the above gives that $g(\zeta) > 0$ for $\zeta > -0.68\gamma - 1.68/\gamma$ and $g(\zeta) < 0$ for $\zeta < -0.905\gamma - 1.905/\gamma$.

**Predicted label for point $(\zeta, x_{2:d+1}^{(t)})$ where $x^{(t)} \in D$ has negative label**

Following the same plan, write our point (denoted by $x$) as $(\zeta, 0, \ldots, 0, 1)$.

*Explicit form for $f$.* Begin by finding

$$\tau_i = \left(\frac{1+\gamma\zeta}{\rho_1\|x\|}\right) \qquad \text{and} \qquad \tau_d = \left(\frac{1-\gamma\zeta}{\rho_2\|x\|}\right).$$

eqn. (10) now gives

$$g(\zeta) = 2^{d-1}a\rho_1\left[\kappa(\tau_0(\zeta)) - \frac{\beta_d\kappa(\tau_d(\zeta))}{\kappa(1)}\right] - \frac{\kappa(\tau_d(\zeta))}{\rho_2\kappa(1)}.$$

Expanding $\kappa(\tau_0(\zeta))$ using Taylor series around $\zeta = -1/\gamma$,

$$\kappa(\tau_0(\zeta)) = \kappa(0) + \kappa'(\tau_0(\theta))\tau_0'(\theta)\left(\zeta + \frac{1}{\gamma}\right),$$

for some $\theta \in [-1, 1]$. For large $d$, $\tau_0(\theta) \approx 0$ and $\tau_0'(\theta) = O(1/\sqrt{d})$. Hence we have

$$\begin{aligned}
g(\zeta) &= \frac{\rho_2\kappa(1) + \rho_1\beta_d}{\xi\rho_1\rho_2}\left[\kappa(0) + O\left(\frac{1}{\sqrt{d}}\right) - \frac{\beta_d\kappa(\tau_d(\zeta))}{\kappa(1)}\right] - \frac{\kappa(\tau_d(\zeta))}{\rho_2\kappa(1)} \\
&= \frac{1}{\rho_2}\left(\frac{\kappa^2(0)}{\xi} - \left(\frac{\kappa^2(0)}{\xi\kappa(1)} + \frac{1}{\kappa(1)}\right)\kappa(\tau_d(\zeta))\right) + o(1).
\end{aligned}$$

As before $g(\zeta) > 0$ whenever the coefficient of $1/\rho_2$ above is bounded above zero which happens for $\zeta \geq 0.73$ (for $\gamma \geq 3$). Similarly, $g(\zeta) < 0$ for $\zeta \leq 0$. $\qquad\square$

# B  Experiments

In this section, we provide experimental details, including hyperparameter tuning setup and some additional experiments.

## B.1  Details on the experimental setting

We will first describe the four datasets that have been used in this work.

1. **Imagenette** (FastAI, 2021): This is a subset of 10 classes of Imagenet, that are comparatively easier to classify.
2. **b-Imagenette**: This is a binarized version of Imagenette, where only a subset of two classes (tench and English springer) is used.
3. **Waterbirds-Landbirds** (Sagawa et al., 2020a): This is a majority-minority group dataset, consisting of waterbirds on water and land background, as well as landbirds on land and water background. This dataset serves as a baseline for checking the dependence of model on the spurious background feature when predicting the bird class, as most of the training examples have waterbirds on water and landbirds on land background.
4. **Imagenet** (Deng et al., 2009): This is the standard benchmark for large scale image classification.
5. **MNIST-CIFAR** (Shah et al., 2020): This is a collage dataset, created by concatenating MNIST and CIFAR images along an axis. This is a synthetic dataset for evaluating the simplicity bias of a trained model.

**Setup**    Throughout the paper, we work with the pretrained representations of the above datasets, obtained by using an Imagenet pretrained Resnet 50. We finetune a 1-hidden layer FCN with a hidden dimension of 100 (8000 for imagenet) on top of these representations (keeping the backbone fixed) using SGD with a momentum of 0.9. Every model is trained for 20000 (100000 for Imagenet) steps with a warmup and cosine decay learning rate scheduler. For each of the runs, we tune the batch size, learning rate and weight decay using validation accuracy. Below are the hyperparameter tuning details:

- Batch size $\in \{128, 256\}$
- Learning rate:
  - Rich regime: $\in \{0.5, 1.0\}$ (for imagenet, $\in \{5.0, 10.0\}$ as learning rate in rich regime needs to scale up with the hidden dimension)
  - Lazy regime: $\in \{0.01, 0.05\}$
- Weight decay: $\in \{0, 1e^{-4}\}$

The final numbers reported are averaged across 3 independent runs with the selected hyperparameters.

**Evaluation**    For Imagenette, b-Imagenette, Imagenet and MNIST-CIFAR, we report the standard test accuracy in all the experiments. For waterbirds, we report train-adjusted test accuracy, as reported in Sagawa et al. (2020a). Precisely, accuracy for each group present in the test data is individually calculated and then weighed by the proportion of the corresponding group in the train dataset.

## B.2  Additional experimental results

In this section, we present a few additional experimental results.

**Results on Imagenet**    The evolution of effective rank of the first layer weight matrix is shown in Figure 5. As can be seen, the weight matrix becomes sufficiently low rank in thre rich regime as the training progresses.

**Singular value decay**    . In Figure 6, we provide the singular value decay of the weight matrix for the first model trained in rich regime. As can be seen, the top few singular values capture most of the Frobenius norm of the matrix.

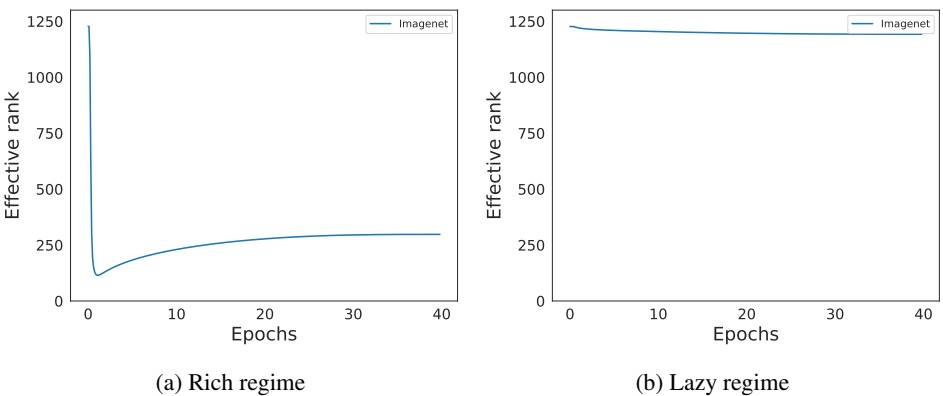

(a) Rich regime           (b) Lazy regime

Figure 5: Evolution of effective rank of first layer weight matrix (dimension - $2048 \times 2000$) for Imagenet dataset in rich and lazy regime.

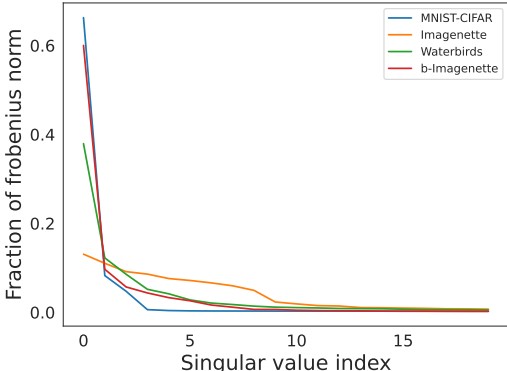

Figure 6: Fraction of Frobenius norm captured by the top $i^{th}$ singular value i.e., $\sigma_i^2 / \sum_{j=1}^{d} \sigma_j^2$ vs $i$ of the first layer weight matrix trained in rich regime for various datasets.

**MNIST-CIFAR**    In Figure 7, we show that an ensemble of $f$ and $f_{\text{proj}}$ has better gaussian robustness than an ensemble of $f$ and $f_{\text{ind}}$ on MNIST-CIFAR dataset.

**Non-linearity of decision boundary**    Figure 8 shows the decision boundary of $f$ and $f_{\text{proj}}$ on 2-dimensional subspace spanned by top two singular vectors of the weight matrix. We observe that the decision boundary of the second model is more non-linear compared to that of the first model. We also report a quantitative measure of non-linearity of the decision boundary along the top two singular vectors for $f$ and $f_{\text{proj}}$. Basically, we fit a linear classifier to the decision boundary and report its accuracy. As shown in Table 6, the test accuracy obtained by the linear classifier for $f_{\text{proj}}$ is less than $f$.

**Variation of LD-SB with depth**    In Figure 9 and 10, we show the evolution of effective rank of weight matrices for depth-2 and 3 ReLU networks. As can be seen, the rank still decreases with training, however the effect is less pronounced for the initial layers. Note that the initialization used in these runs was the feature learning initialization as proposed in Yang & Hu (2021).

**Comparison of Mistake diversity with Evading SB**    Table 7 shows the results of comparing $OrthoP$ with Evading-SB (Teney et al., 2022), with regards to diversity metrics. As can be seen, $OrthoP$ is either better or comparable to Evading-SB.

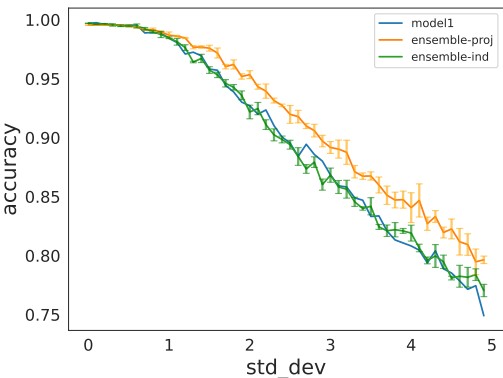

Figure 7: Variation of test accuracy with the standard deviation of Gaussian noise added to the pretrained representations of MNIST-CIFAR dataset. Model 1 is kept fixed, and values for both the ensembles are averaged across 3 runs.

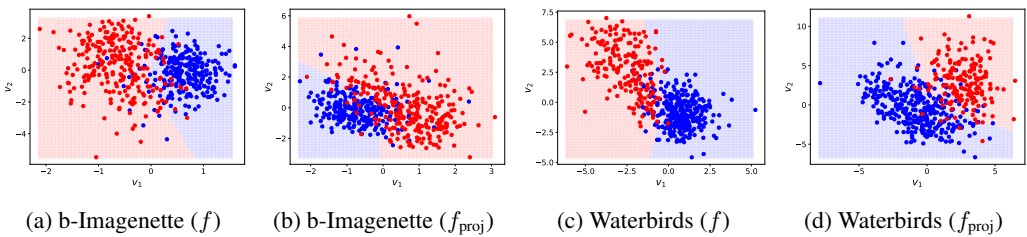

(a) b-Imagenette ($f$)  (b) b-Imagenette ($f_{\text{proj}}$)  (c) Waterbirds ($f$)  (d) Waterbirds ($f_{\text{proj}}$)

Figure 8: Decision boundaries for $f$ and $f_{\text{proj}}$ for B-Imagenette and Waterbirds datasets, visualized in the top 2 singular directions of the first layer weight matrix. The decision boundary of $f_{\text{proj}}$ is more non-linear compared to that of $f$.

Table 6: Quantitative measurement of non-linearity of decision boundary – accuracy of fitted linear classifier to the decision boundary

| Dataset | Linear-Classifier-Acc($f$) | Linear-Classifier-Acc($f_{\text{proj}}$) |
|---|---|---|
| b-Imagenette | 96.12 | $95.28 \pm 0.2$ |
| Waterbirds | 97.28 | $93.24 \pm 0.24$ |

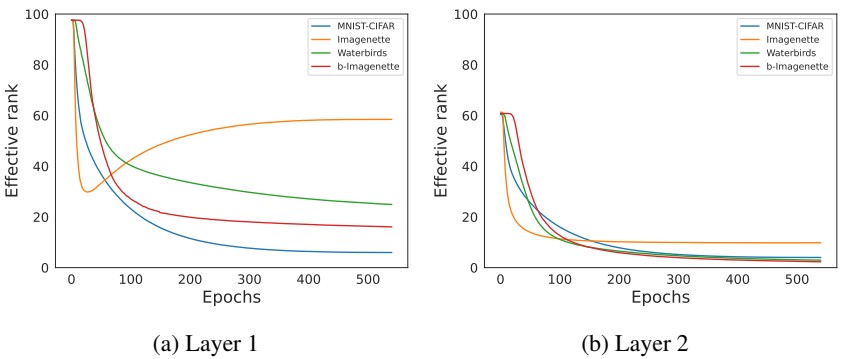

(a) Layer 1  (b) Layer 2

Figure 9: Evolution of effective rank of the weight matrices for a depth-2 ReLU network on Resnet-50 pretrained representations of the dataset

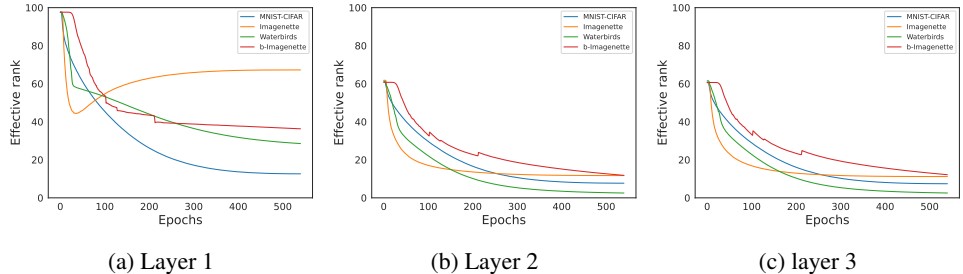

|     (a) Layer 1     |     (b) Layer 2     |     (c) layer 3     |

Figure 10: Evolution of effective rank of the weight matrices for a depth-3 ReLU network on Resnet-50 pretrained representations of the dataset

Table 7: Mistake diversity and class conditioned logit correlation of models trained sequentially after projecting out important features of the first model (Mist-Div $(f, f_{\text{proj}})$ and CC-LogitCorr $(f, f_{\text{proj}})$ resp.) and using the method mentioned in Evading-SB ((Mist-Div $(f, f_{\text{esb}})$ and CC-LogitCorr $(f, f_{\text{esb}})$ resp.). Numbers have been bolded if they are better than the other even after taking one standard deviation into account.

| Dataset | Mist-Div $(f, f_{\text{esb}})$ $(\uparrow)$ | Mist-Div $(f, f_{\text{proj}})$ $(\uparrow)$ | CC-LogitCorr $(f, f_{\text{esb}})$ $(\downarrow)$ | CC-LogitCorr $(f, f_{\text{proj}})$ $(\downarrow)$ |
|---|---|---|---|---|
| B-Imagenette | $17.86 \pm 0.0$ | $\mathbf{21.15 \pm 1.57}$ | $90.61 \pm 0.48$ | $90.86 \pm 1.08$ |
| Imagenette | $11.23 \pm 0.24$ | $11.44 \pm 0.65$ | $92.48 \pm 0.68$ | $\mathbf{91 \pm 0.59}$ |
| Waterbirds | $13.6 \pm 0.36$ | $\mathbf{14.53 \pm 0.48}$ | $95.41 \pm 0.41$ | $\mathbf{93.81 \pm 0.48}$ |
| MNIST-CIFAR | $0.0 \pm 0.0$ | $5.56 \pm 7.89$ | $76.19 \pm 2.38$ | $78.74 \pm 2.28$ |
| Imagenet | $4.95 \pm 0.11$ | $\mathbf{12.31 \pm 0.16}$ | $98.83 \pm 0.0$ | $\mathbf{92.52 \pm 0.01}$ |

# C  Extended Related Works

In this section, we provide an extensive literature survey of various topics that the paper is based on.

**Low rank Simplicity Bias in Linear Networks**  Multiple works have established low rank simplicity bias for gradient descent on linear networks, both for squared loss as well as cross-entropy loss. For squared loss, Gunasekar et al. (2017) conjectured that the network is biased towards finding minimum nuclear norm solutions for two-layer linear networks. Arora et al. (2019) refuted the conjecture and instead argued that the network is biased towards finding low rank solutions. Razin & Cohen (2020) provided empirical support to the low rank conjecture, by providing synthetic examples where the network drives nuclear norm to infinity, but minimizes the rank of the effective linear mapping. Li et al. (2021) established that for small enough initialization, gradient flow on linear networks follows greedy low-rank learning trajectory. For binary classification on linearly separable data, Ji & Telgarsky (2019) showed that the weight matrices of a linear network eventually become rank-1 as training progresses.

**Low rank Simplicity Bias in Non-Linear Networks**  There have been a few works establishing low-rank simplicity bias for non-linear networks. Phuong & Lampert (2021) showed that, for orthogonally separable datasets, the rank of first layer weights for two-layer ReLU networks eventually tends to two under gradient flow on logistic loss. Frei et al. (2023) extend this result to nearly-orthogonal datasets and Leaky-ReLU networks. Timor et al. (2023) showed that, maximum margin ReLU networks of sufficiently large depth implicitly also have low rank. Huh et al. (2021) empirically established that the rank of the embeddings learnt by a neural network with ReLU activations goes down as training progresses. Galanti & Poggio (2022) provided an intuition behind the relation between the rank of the weight matrices and various hyperparameter such as batch size, weight decay etc. In contrast to these works, for 1 layer nets, we theoretically and empirically establish that the network depends on an extremely low dimensional projection of the input, and this bias can be utilized to develop a robust classifier.

**Adversarial Robustness**  Some of the recent works have also shown that gradient flow on shallow networks generally leads to non-robust classifiers. Vardi et al. (2022) showed that for nearly orthogonal datasets, gradient flow on 2-layer ReLU networks converges to non-robust networks, even when robust classifiers are possible for the dataset. Melamed et al. (2023) show similar results for datasets which lie on a low-dimensional linear subspace.

**Relation to OOD**  Many recent works in OOD detection (Cook et al., 2020; Zaeemzadeh et al., 2021) explicitly create low-rank embeddings so that it is easier to discriminate them for an OOD point. Other works also implicitly rely on the low-rank nature of the embeddings. Ndiour et al. (2020) use PCA on the learnt features, and only model the likelihood along the small subspace spanned by the top few directions. Wang et al. (2022) utilise the low rank nature of the embeddings to estimate the perpendicular projection of a given data point to this low rank subspace and combine it with logit information to detect OOD datapoints. While there have been works implicitly utilizing the low rank property of embeddings, we note that our paper (i) demonstrates low rank property of the *weights*, rather than that of embeddings, and (ii) shows that it is a consequence of SB.

**Other Simplicity Bias**  There have been many works exploring the nature of simplicity bias in neural networks, both empirically and theoretically. Kalimeris et al. (2019) empirically demonstrated that SGD on neural networks gradually learns functions of increasing complexity. Rahaman et al. (2018) empirically demonstrated that neural networks tend to learn lower frequency functions first. Ronen et al. (2019) theoretically established that in NTK regime, the convergence rate depends on the eigenvalues of the kernel spectrum. Hacohen et al. (2020) showed that neural networks always learn train and test examples almost in the same order, irrespective of the architecture. Pezeshki et al. (2021) proposes that *gradient starvation* at the beginning of training is a potential reason for SB in the lazy/NTK regime but the conditions are hard to interpret. In contrast, our results are shown for any dataset in the IFM model in the *rich* regime of training.  Lyu et al. (2021) consider anti-symmetric datasets and show that single hidden layer input homogeneous networks (i.e., without *bias* parameters) converge to linear classifiers. However, such networks have strictly weaker expressive power compared to those with bias parameters. Hacohen & Weinshall (2022) showed that for deep linear networks, in NTK regime, they learn the higher principal components of the input data first.

Most of the previous works used simplicity bias as a reason behind better generalization of neural nets. However, Shah et al. (2020) showed that extreme simplicity bias could also lead to worse OOD performance.

**Learning diverse classifiers**: There have been several works that attempt to learn diverse classifiers. Most works try to learn such models by ensuring that the input gradients of these models do not align (Ross & Doshi-Velez, 2018; Teney et al., 2022). Xu et al. (2022) proposes a way to learn diverse/orthogonal classifiers under the assumption that a complete classifier, that uses all features is available, and demonstrates its utility for various downstream tasks such as style transfer. Lee et al. (2022) learns diverse classifiers by enforcing diversity on unlabeled target data.

**Spurious correlations**: There has been a large body of work which identifies the reasons for spurious correlations in NNs (Sagawa et al., 2020b) as well as proposing algorithmic fixes in different settings (Liu et al., 2021; Chen et al., 2020b).

**Implicit bias of gradient descent**: There is also a large body of work understanding the implicit bias of gradient descent dynamics. Most of these works are for standard linear (Ji & Telgarsky, 2019) or deep linear networks (Soudry et al., 2018; Gunasekar et al., 2018). For nonlinear neural networks, one of the well-known results is for the case of 1-hidden layer neural networks with homogeneous activation functions (Chizat & Bach, 2020), which we crucially use in our proofs.

## D    More discussion on the extension of results to deep nets

Extending our theoretical results to deep nets is a very exciting and challenging research direction. For shallow as well as deep nets, even in the mean field regime of training, results regarding convergence to global minima have been established (Chizat & Bach, 2018; Fang et al., 2021). However, to the best of our knowledge, only for 1-hidden layer FCN (Chizat & Bach, 2020), a precise characterization of the global minima to which gradient flow converges has been established. Understanding this implicit bias of gradient flow is still an open problem for deep nets, which we think is essential for extension of our results to deep nets.

## E    Convergence to $\mathcal{F}_1-$max-margin classifier for ReLU networks

In this section, we will provide a brief background on Wasserstein gradient flow and state the precise result of Chizat & Bach (2020) regarding the asymptotic convergence point of gradient flow on ReLU networks. We will follow the notation of Chizat & Bach (2020) for ease of the reader. In this entire section, we will consider that a neural network is parameterized by a probability measure $\mu$ on the neurons and is given by

$$h(\mu, x) = \int \phi(w, x) d\mu(w)$$

where $\phi(w, x) = b(a^\top(x, 1))_+$ (+ denotes the positive component, i.e the ReLU activation) with $w = (a, b) \in \mathbb{R}^{d+2}$.

### E.1    Wasserstein gradient flow

Gradient flow can be defined for many functions $f$ over a general metric space $\mathcal{X}$. For a given step size $\eta$, define

$$x_{k+1} \in \arg\min f(x) + \frac{1}{2\eta} d(x, x_k)^2$$

where $d$ is the metric associated with $\mathcal{X}$. With appropriate interpolation schemes (Santambrogio, 2016), this curve converges to the gradient flow curve as step size tends to 0.

Wasserstein metric on the space of probability measures is defined as

$$W_p(\nu_1, \nu_2) = \inf_{\gamma \in \Gamma_{\nu_1, \nu_2}} \left[ \int \|x - y\|^p d\gamma \right]^{1/p}$$

where $\nu_1, \nu_2$ are two probability measures and $\gamma$ is a coupling between them (i.e marginals of $\gamma$ are $\nu_1$ and $\nu_2$). Here, we will be particularly concerned with the case $p = 2$. For two discrete measures

$\nu_1 = \frac{1}{m} \sum \delta_{x_i}$ and $\nu_2 = \frac{1}{m} \sum \delta_{y_j}$, their Wasserstein distance is defined as

$$W_2^2(\nu_1, \nu_2) = \frac{1}{m} \min \|x_i - y_{\sigma(i)}\|^2$$

over all permutations $\sigma : \{1, .., m\} \to \{1, ..., m\}$. Notice that if we are considering a small neighborhood of $\nu_1$, then the mapping would remain the same within that small neighborhood. Thus, Wasserstein gradient flow on the discrete measure would be the same as gradient flow on the particles as the notion of distance is the same locally. This intuition leads to the proof that gradient flow on a 2-layer neural net converges to Wasserstein gradient flow on the probability measure $\mu$ as width tends to infinity. This is made formal in Chizat & Bach (2018).

### E.2 Asymptotic convergence point of gradient flow on ReLU networks

A neural network is parameterized by a probability measure $\mu$ on the neurons and is given by

$$h(\mu, x) = \int \phi(w, x) d\mu(w)$$

where $\phi(w, x) = b(a^\top(x, 1))_+$ (+ denotes the positive component, i.e the ReLU activation) with $w = (a, b) \in \mathbb{R}^{d+2}$. As the network is 2-homogeneous, a projection of the measure $\mu$ on the unit sphere can be defined. The projection operator $(\Pi_2)$ on the sphere for a measure $\mu$ is defined such that for any continuous function $\varphi$ on the sphere,

$$\int_{\mathbb{S}^{d+1}} \varphi(\theta) d[\Pi_2(\mu)](\theta) = \int_{\mathbb{R}^{d+2}} \|w\|^2 \varphi(w/\|w\|) d\mu(w)$$

Now, let $\rho$ denote the input distribution on the input space $\mathcal{X}$ and let the labeling function $y : \mathcal{X} \to \mathcal{Y}$ be deterministic. Then, consider the population objective given by

$$F(\mu) = -\log \left[ \int_{\mathcal{X}} \exp(-y(x)h(\mu, x)) d\rho(x) \right]$$

Note that $\log$ doesn't affect the direction of the gradients, thus, the trajectory of gradient flow on this loss is the same as on exponential loss. Also, let the population smooth margin be given by

$$S(f) = -\log \left( \int_{\mathcal{X}} \exp(-f(x)) d\rho(x) \right)$$

For this particular case, $f(x) = y(x)h(\mu, x)$. Denote $y(x) \cdot h(\mu, x)$ by $\hat{h}(\mu)$.

**Theorem E.1.** *Suppose that $\rho$ has bounded density and bounded support, and labeling function $y$ is continuous, then there exists a Wasserstein gradient flow $(\mu_t)$ on $F$ with $\mu_0 = \mathcal{U}(\mathbb{S}^d) \otimes \mathcal{U}\{-1, 1\}$, i.e, input (resp. output) weights uniformly distributed on the sphere (resp. on $\{-1, 1\}$). If $\nabla S(\hat{h}(\mu_t))$ converges weakly in $\mathcal{P}(\mathcal{X})$, if $\bar{\nu}_t = \Pi_2(\mu_t)/([\Pi_2(\mu_t)](\mathbb{S}^{d+1}))$ converges weakly in $\mathcal{P}(\mathbb{S}^{d+1})$ and $F'(\mu_t)$ converges in $C^1_{loc}$ to $F'$ that satisfies the Morse-Sard property, then $h(\bar{\nu}_\infty, .)$ is a maximizer for $\max_{\|f\|_{\mathcal{F}_1} \leq 1} \min_{x \in \mathcal{X}} y(x)f(x)$.*

where $\mathcal{P}(\mathcal{X})$ denotes the space of probability distributions on $\mathcal{X}$ and $[\Pi_2(\mu_t)](\mathbb{S}^{d+1})$ denotes the total mass of the measure $\Pi_2(\mu_t)$ on $\mathbb{S}^{d+1}$.

To parse the theorem, note that

$$\nabla S(f) = \frac{\exp(-f(x)) d\rho(x)}{\int_{\mathcal{X}} \exp(-f(x')) d\rho(x')}$$

Thus, $\nabla S(f)$ convergence means that the exponentiated normalized margins converge. Also, $\bar{\nu}_t$ is similar to the directional convergence of weights, however, in this case, weights are replaced by directions in $\mathbb{S}^{d+1}$. For explanation of the Morse-Sard property and the metric $C^1_{loc}$, please refer to Appendix H of Chizat & Bach (2020).

