# OpenReview forum: "Simplicity Bias in 1-Hidden Layer Neural Networks"
_NeurIPS.cc/2023/Conference — NeurIPS 2023 poster_

### Official Review · Reviewer_W8La · 2023-06-13

**Soundness:** 2 fair
**Presentation:** 2 fair
**Contribution:** 3 good
**Rating:** 5
**Confidence:** 4

**Summary:**

The concept of simplicity bias (SB), that is, the behaviour for machine learning models only to learn the simplest features at hands when solving a given task, even when there are more complex but more robust features available, is central in this work. First, the authors aim toward a mathematical definition of a special case of it : low dimensional input dependence simplicity bias. The authors study SB in the special case of 1-hidden layer neural networks; from a theoretical perspective, with infinite width, and from a practical point of view, with finite width. The author propose an algorithm, OrthoP, an ensemble method for obtain diverse classifiers.

**Strengths:**

1. The related work section is thorough and discusses many topics, especially considering the work that has been made in Section C (supplementary material). I feel like such an aspect is truly important in a conference paper and is too oftenly neglected.

2. The idea of having « independent », or orthogonal, representations of the input of a problem in order to obtain diverse classifiers is an interesting idea and is intuitive.

**Weaknesses:**

_Major_

1. The main result of the article holds for a very specific case : on an IFM dataset (binary classification, with a few strong assumptions on it), which is tackled by a 1-hidden layer neural network, in the case of infinite width, with ReLU activation function, with cross-entropy loss. If the assumptions concerning the predictor are defended by the argument that this is respected when doing, for example, transfer learning, or fine-tuning a large model that has already been trained, not much argument is given toward the IFM assumption, apart from the fact that it « has a long history and is widely studied » (Lines 71-72).

2. The metrics described in subsection 4.4.1. ($P_{\perp}-LC$, $P-LC$, $P_{\perp}-pC$ and $P-pC$) do not consider the correlation between examples $x_1$ and $x_2$, or the label values $y_1$ and $y_2$. That is a problem, because for example, for P-LC and P-pC on their current form, the goal would not be to obtain the highest values possible (as currently proposed), but to obtain pred($f(\tilde{x})$) = pred($f(x_2)$) when $y_1$ = $y_2$, and pred($f(\tilde{x})$) $\neq$ pred($f(x_2)$) when $y_1 \neq y_2$ for P-pC and to have , ||f($\tilde{x}$) – f(x_2)|| / ||f(x_2)|| being low when $x_1$ and $x_2$ are similar, for example, for P-LC. Therefore, I think those metric are too simplistic and might not reflect what their purpose was intended to be.

3. The article is dense. It seems like, in order to save place, many lines of relevant discussion have been removed. For examples, not a line of discussion regarding the results from Table 1, 2, 3 and 4 are given, while they constitute a big part of the empirical experiments. This leaves a feeling of lack in justifications, or the need for the reader to decide whether the reported results are interesting or not.

4. The experimentations given in section 5 do not convince me. For example, Table 5 report mistake diversity when using OrthoP versus when using independent training of two models. It therefore is not surprising that OrthoP yields more diverse predictor when compared to a method that simply does not aim for that goal. The fact that no comparison is made with methods for obtaining diverse classifier in ensemble methods makes it that the reported results are hard to interpret (same goes for robustness experiments). Are the difference between OrthoP and the vanilla approach big or not? The metrics that are considered do not help in this sense. (The same goes for Table 1 and Table 2, but is worse in this last case for it is hard to interpret the results only on a single approach; should we compare P$_\perp$-LC and P-LC together? And P$_\perp$-pC and P-pC together?) The use of 5 different datasets satisfies me, but as for the robustness experiment, I do not feel like only looking at Gaussian perturbation of images yields a solid argument toward what’s defended.



_Typos / Minor_

1. The definition of LD-SB seems a bit arbitrary, for there are no criteria to determine how k should be small relatively to d, and there are no relationship between $\epsilon_1$ and $\epsilon_2$.

2. In the definition of LD-SB, I would state clearly that model g and model f must have the same architecture, for it is implicit but crucial. In the same line of thoughts, it is mentioned that model g is trained on examples $(P_{\perp}x,y)$, where $(x,y) \~ \mathcal{D}$, but it is not stated whether the accuracy (Acc(.)) refers to the training accuracy or the test accuracy.

3. Line 72 : « IFM has a long history and is widely studied », but no citation toward recent or any other important work than Lewis (1998) is given, so the statement that IFM is currently widely studied is not supported.

4. On Line 145 : « Consider instances $x \in \mathcal{R}^d$ […] » instead of $\mathbb{R}^d$.

5. On Lines 147-148, the model is defined as a function of weights and biases for the hidden layer, but only weights for the output layer. Was it on purpose?

6. In subsection 3.2, the parameters from the hidden layer are sampled independently from the parameters from the output layer. But later in subsection 3.3, we are told that they are sampled dependently. Does that have something to do with the infinite width regime, or was that a mistake?

7. At the beginning of Line 213 : missing end-of-sentence point.

8. Line 306 : « […] it turns out that the \textbf{effective} rank [...] ».

9. Line 323 : no closed parenthesis in denominator

10. Line 324 : « […] where we abuse notation by using f($x_i$) [...] »; no such notation is used.

11. The only place where a reported accuracy is explicitly defined as being test accuracy is for Figure 4. It might be implicit that the same goes for Tables 3 and 4, but I would make it explicit.

**Questions:**

1. How would you reconcile simplicity bias with the widely used technique of regularising in order to obtain simpler models and therefore being more robust? The two approaches seems contradictory. Especially since the use of weight decay was considered in the experimental section, which leads to simpler models.

2. I might be wrong here, but it seems like for a projection matrix $P$, there exist an infinite number of orthogonal subspaces $P_{\perp} \in \mathbb{R}^{d \times d}$, and especially if rank(P) $\neq$ d, for there are now more degrees of freedom for constructing $P_{\perp}$ . Once a projection matrix $P$ is found, how is $P_{\perp}$ found / computed?

3. How LD-SB is a special case of SB (as stated on Line 50)? The definition of SB given in the abstract goes as follow : « […] [To] learn only the simplest features to solve a task at hand, even in the presence of other, more robust but more complex features ». Isn’t that what LD-SB aims to verify? When a predictor only uses a handful of features for rendering predictions while there are other that would be useful?

4. Line 225 : « Though we assume axis aligned features, our results also hold for any rotation of the dataset. » Is there a proof of that somewhere in the article?

5. In Theorem 4.1, what does $\delta$ refers to? It is mentioned twice on Line 234, and nowhere else in the core of the paper. It might be explained in the proof, but since the proof is in supplementary material, it would be necessary to explain it in a few words.

6. Is Theorem 4.2 an original result? If not, where does it come from?

7. Line 299 : « In [the rich regime], we set P to be the subspace spanned by the top singular directions of the first layer weight matrix. » Why this choice of P? A justification would be necessary in the article.

8. On Lines 309 to 311, an optimisation problem is described for obtaining P in the lazy regime. The optimisation of a d x d matrix, especially when d is high, can lead to an optimisation of tens of thousands of parameters simultaneously. How well did it converge? How much computing time was necessary in order to obtain an acceptable P matrix?

9. When considering OrthoP in order to obtain a diverse ensemble predictor, is the ensemble necessarily constituted of two predictors? It would have been nice to witness a technique involving the adding of « uncorrelated » predictors until a certain criteria is met.

10. I am not familiar with the metrics described in Section 4.4.1 and those used in Section 5; do they appear in the literature? Are they commonly used?

**Limitations:**

I feel like the authors discussed adequately the limitations of their work.

---

> ### Author Rebuttal · Authors · 2023-08-09
>
> We thank the reviewer for their valuable feedback.
>
> 1. **IFM assumption** - The IFM assumption is a simple and intuitive assumption that enables us to define features – in this model features are distinguished from each other through conditional independence. IFM played a key role in developing a thorough understanding of naive Bayes classifiers e.g., [1, 2] which has since been extended to more general data models [3, 4]. Similarly, given the close connection between features and simplicity bias (SB), we believe that IFM is a canonical setting to study SB.
>
>     *Linear separability*: Using linear evaluation protocol is the standard practice in many domains e.g. self-supervised learning domain. The pretrained representations obtained by self-supervised learning are clustered enough that they can be well-separated using a linear classifier [5]. Moreover, as demonstrated in Figure 8 and Table 6 in the Appendix, the pretrained representations on various datasets are close to being linearly separable.
>
> 2. **Metrics in subsection 4.4.1** - We are not sure we fully understand the reviewer’s question. The main reasoning behind using these metrics is to verify the points specified in the definition of LD-SB (Definition 1.1), specifically the second point. In particular, $P_\perp-PC$ exactly quantifies the value of $\epsilon_1$ in Definition 1.1. Moreover, a very similar metric to these, randomized accuracy, was used for measuring the dependence of a classifier on a subset of coordinates in [6]. The mistake diversity used in Table 5 is a general metric for checking functional similarity of neural networks [7] .
>
> 3. **Discussion of results** - We would like to point out that in Section 4.4.1, we had already discussed the relevance of various metrics being displayed in Tables 1, 2, 3 and 4. However, in the updated version, we will make sure to have more discussion regarding the results in their respective sections.
>
> 4. **Experiments in Section 5** - Based on the reviewer’s suggestion, we conducted comparison experiments with another diverse training method  - Evading SB [8]. The results are attached in the one page pdf. Our method generally performs comparable to or better than this method in terms of class conditioned logit correlation or mistake diversity. In terms of gaussian robustness, the two methods are comparable as well, with our method slightly outperforming on ImageNet. Regarding gaussian robustness, we consider it as a metric that estimates the input space margin of the model.
>
> 5. **Simplicity Bias and Regularization** - Our paper indeed shows that in some cases, simplicity bias does not lead to robust classifiers. This was first pointed out explicitly in [6], and we are demonstrating it in a more general setting. The use of weight decay in the experimental section is to replicate practical setup and demonstrate the adverse effects of simplicity bias in the setting. Also, note that F1 max margin classifier is the global minimizer of a limiting case of $\ell_2$ regularization, where the weight decay coefficient tends to zero [9]. Our results prove that F1 max margin classifier suffers from the adverse effects of simplicity bias.
>
> 6. **Construction of $P_{\perp}$** - This is not the case. For a given projection matrix $P$,  $P_{\perp} = I - P$.
>
> 7. **LD-SB vs SB** - LD-SB provides a specific way of instantiating SB. Its not necessary that SB is a linear subspace of the input features, it could be a lower dimensional non-linear manifold. However, for 1 hidden layer networks, because of the low dimensional structure of the weights itself, we can identify the simple classifier within a low dimensional linear subspace of the inputs.
>
> 8. **Proof for rotation of dataset** - We will add this proof to the appendix. It follows as we can rotate the weight matrix in response to any rotation of the inputs.
>
> 9. **Theorem 4.1** - Delta represents the dirac delta distribution. Thanks for the catch.
>
> 10. **Theorem 4.2** - Yes, theorem 4.2 is an original result. Its proof is provided in Appendix A.2
>
> 11. **Choice of P** -  We have chosen this $P$ as the weight matrix itself is low rank. One justification is provided by Theorem 4.1, which shows that for a linearly separable dataset within IFM, this indeed represents the direction of the linearly separable coordinate. Thus projecting out this direction will force the model to rely on non-linearly separable coordinates.
>
> 12. **Optimization of P in lazy regime** - If the projection is done over a $k$ dimensional subspace (where k is generally in the range of 1-10 as mentioned in Table 1), then the optimization is done over $d \times k$ basis matrix, which can be used to create the projection matrix. This generally converged well, and was completed within a minute.
>
> We hope that we addressed the reviewers questions and concerns and would appreciate it if the reviewer would update their score accordingly.
>
> [1] An empirical study of the naive Bayes classifier by I Rish -- IJCAI 2001 workshop on empirical methods in artificial intelligence, 2001
>
> [2] Naive (Bayes) at forty: The independence assumption in information retrieval DD Lewis - ECML, 1998
>
> [3] An improvement to naive bayes for text classification W Zhang, F Gao - Procedia Engineering, 2011
>
> [4] A non-parametric mixture of Gaussian naive Bayes classifiers based on local independent features AH Jahromi, M Taheri - AISP 2017
>
> [5] Chen et al. A simple framework for contrastive learning of visual representations. In ICML, pp. 1597–1607. PMLR, 2020a.
>
> [6] Shah et al. - The pitfalls of simplicity bias in neural networks. Advances in NeurIPS, 33:9573–9585, 2020.
>
> [7] Klabunde at al. 2023 - Similarity of Neural Network Models, arXiv:2305.06329
>
> [8] Teney et al. 2021 - Evading the Simplicity Bias: Training a Diverse Set of Models Discovers Solutions with Superior OOD Generalization, CVPR, 2022
>
> [9] Lenaic Chizat 2020 - Gradient descent for wide two-layer neural networks – II: Generalization and implicit bias

---

> > ### Comment · Reviewer_W8La · 2023-08-11
> > **Rebuttal - Response**
> >
> > 1. IFM assumption - In the presented approach, not only does the IFM assumption is made, but the linear separability (plus other assumptions, discussed in 4.1) as well. It is the combination of those two frameworks that leads to one truly specific case.
> >
> > "The pretrained representations obtained by self-supervised learning are clustered enough that they can be well-separated using a linear classifier". But if we consider the representations to be the features that are linearly separable, then can we assume that those same features are independent, knowing for a fact that they are a transformation of the inputs of the problem?
> >
> > If we consider the representations to be the feature of interest, before stating whether or not k << d (for example, in Table 1), since d does not correspond to the number of inputs in the problem, we would need to know the size of the representations.
> >
> > 2. Metrics in subsection 4.4.1 - I think the metrics, in their current form, ignore important information. For example: it is written in Tables 1 and 2 that P-pC should be as high as possible. Thus, we want pred($f(\tilde{x})$) $\neq$ pred($f(x_2)$). But shouldn't this metric consider if the values $y_1$ and $y_2$ are different or similar? Let's say $x_1$ and $x_2$ are rather similar and $y_1$ = $y_2$, then why would we want pred($f(\tilde{x})$) $\neq$ pred($f(x_2)$)? I feel like ignoring those information makes it hard to determine how P$_\perp$ is handled by f and makes the overall information provided by P-pC (and P-LC, for the same reasons) doubtful.
> >
> > 3. Discussion of results - Discussing the relevance of various metrics being displayed in Tables 1, 2, 3 and 4 cannot replace discussing the quality of the obtained results. Glad to hear words on that matter will be added.
> >
> > I'll be waiting for the few other clarifications I need before adjusting my score.

---

> > > ### Author Response · Authors · 2023-08-17
> > >
> > > Thanks for the response.
> > >
> > > 1. **IFM Assumption** -  It is indeed correct that the pretrained representations obtained need not have independent features in general. However in this case, since the pretraining task (full Imagenet classification) requires several features, and independent features lead to better downstream performance [1], it is likely that the features are indeed independent. In fact, we have computed the correlation matrix of pretrained representations and observed that, for all datasets except MNIST-CIFAR, most pairs of features (>95%) have (normalized) correlation less than 0.3. Because of the significant difference with the pretraining dataset (Imagenet), we think that a notable fraction of the neurons are correlated for MNIST-CIFAR.     Moreover, there are also theoretical works which show that as networks get wider, neurons in a hidden layer tend to become independent [2,3]. Below is the table showing fraction of feature pairs having correlation less than a particular quantity.
> > >
> > >     | Dataset | $< 0.1$ | $< 0.3$ | $<0.5$ | $<0.7$ |
> > >     |----------|----------|-----------|---------|---------|
> > >     | Imagenette | $0.66$ | $0.95$ | $0.99$ | $1.0$ |
> > >     | B-Imagenette | $0.7$ | $0.95$ | $0.99$ | $1.0$ |
> > >     | Waterbirds | $0.68$ | $0.98$ | $1.0$ | $1.0$ |
> > >     | Imagenet | $0.74$ | $0.99$ | $1.0$ | $1.0$ |
> > >     | MNIST-CIFAR | $0.34$ | $0.37$ | $0.38$ | $0.38$ |
> > >
> > >     Regarding the size of the representations, this is equal to 2048 (this is provided in the Appendix, but will also mention it in the main text). Compared to this, the largest rank used in Tables 1 and 2 is 200 (< 10%).
> > >
> > >     [1] - Zbontar, Jure, et al. "Barlow twins: Self-supervised learning via redundancy reduction." International Conference on Machine Learning. PMLR, 2021.
> > >
> > >     [2] - Yang, Greg, and Edward J. Hu. "Tensor programs iv: Feature learning in infinite-width neural networks." International Conference on Machine Learning. PMLR, 2021.
> > >
> > >     [3] - Bordelon, B. and Pehlevan, C. “Self-consistent dynamical field theory of kernel evolution in wide neural networks.” - Advances in Neural Information Processing Systems, 2022
> > >
> > > 2. **Metrics in subsection 4.4.1** - Thanks for the explanation, We do understand this issue with $P-pC$ and $P-LC$. One way to tackle this issue is to randomize only across (not within) classes, i.e, $y_1 \neq y_2$. We indeed tried this and obtained very similar trends to what we presented in the paper (below are the obtained values for the rich regime  - Table 1 in the paper).
> > >
> > >     | Dataset | $P_{\perp}-LC$ | $P-LC$ | $P_{\perp}-pC$ | $P-pC$ |
> > >     |----------|----------------|--------|-----------------|--------|
> > >     |B-Imagenette| $32.08 \pm 0.65$ | $165.41 \pm 0.28$ | $6.61 \pm 0.22$ | $81.97 \pm 0.31$ |
> > >     |Imagenette|$33.47 \pm 0.49$| $148.86 \pm 0.3$ | $11.97 \pm 0.44$ | $96.2 \pm 0.07$ |
> > >     |Waterbirds| $25.08 \pm 0.55$| $128.22 \pm 0.98$ | $6.66 \pm 0.38$ | $46.39 \pm 0.44$ |
> > >     |MNIST-CIFAR| $51.53 \pm 2.99$| $186.94 \pm 0.57$| $11.05 \pm 1.07$| $89.24 \pm 1.23$ |
> > >     |Imagenet| $15.78 \pm 0.05$ | $132.14 \pm 0.05$ | $13.27 \pm 0.04$ | $99.78 \pm 0.01$ |

---

> > > > ### Comment · Reviewer_W8La · 2023-08-20
> > > >
> > > > I am still concerned about the interpretation of Table 1. The fact is, there is no baseline to compare to (Weaknesses - Major - 4). Therefore, it is hard to tell how much the results are encouraging. (I just realised: it should be mentioned explicitly that for P-pC and $P_{\perp}$-pC, the number displayed are percentages.) Also, I still doubt the theoretical foundation and the soundness of the definition given to LD-SB in Definition 1.1 (Weaknesses - Minor - 1);  for example, the points we have discussed so far concerning taking into account the similarity between $x_1$ and $x_2$ and the labels in the Point 2 of Definition 1.1. Concerning Point 3, considering that the projection matrix onto the subspace orthogonal to P is computed as $P_{\perp}$ = I - P, I don't intuitively see why there shouldn't exist a matrix of weights, for the first hidden layer of g that leads to a predictor similar to f. In other words, there might exist a simple way to parameterize g such that the predictions, given $P_{\perp}$x is similar to the prediction of f given Px. Nevertheless, a step forward is made in understanding the simplicity bias involving neural networks in a particular setup. In light of our discussions, and taking into account that the clarifications given by the authors to many of my questions will be clarified/put in the final version of the manuscript, I now lean toward borderline acceptance; I'll adjust my score accordingly.

---

> > > > > ### Author Response · Authors · 2023-08-21
> > > > >
> > > > > Thanks for updating the score, and for the additional comments. Regarding Table 1, we would like to point out that we did a comparison with another diverse ensemble method named Evading-SB. We have the results attached in the one page pdf of the rebuttal phase, where the numbers for OrthoP seem comparable or slightly better, while being theoretically grounded.
> > > > >
> > > > > Regarding whether it is necessary for a $g$ to exist which gets the same features as ones obtained by $f$, consider the linearly separable IFM dataset. By theorem 4.1, the top singular vector of $f$ highlights the linearly separable coordinate. So, once we project out this coordinate (by doing I-P, we essentially set this coordinate to 0 for every point, in other words, the matrix I-P is not invertible), $g$ has to rely on other non-linearly separable coordinates for classifying the dataset. Thus, if features are not repeated in the dataset (which as argued above is less likely due to the hard pretraining task, and as evidenced by the small correlation across features), it is not possible for the network to use the features that have been discarded by projecting them out. We will add this discussion to the paper to clarify these arguments.

---

### Official Review · Reviewer_gAuS · 2023-06-20

**Soundness:** 3 good
**Presentation:** 3 good
**Contribution:** 3 good
**Rating:** 6
**Confidence:** 4

**Summary:**

The main theoretical result in the paper is an application of the theorem of Chizat and Bach (COLT 2020), who showed implicit bias to margin maximisation with respect to the variation norm for logistic loss gradient flow on infinite-width one-hidden layer homogeneous networks.  The focus of the result is when the network activation is ReLU and the training data is given by an independent features model, in which only one feature is linearly separable and the data without that feature is separable (however not linearly); the result then establishes that there is a unique maximum-margin classifier, which only uses the linearly separable feature.  The other theoretical result in the paper is similar; in contrast to the first which applies for initialisation scales that result in the rich regime, it focuses on the lazy regime and on a more streamlined point-mass independent features model for the training data.  In the experimental part, a one-hidden layer ReLU network is trained on top of a pretrained Resnet-50 network with 2048 features, in both the rich and the lazy regimes, and on several datasets (four real: Imagenette, b-Imagenette, waterbirds-landbirds, and Imagenet; and one designed: MNIST-CIFAR).  The experimental results support the hypothesis of simplicity bias, namely that the training converges to a classifier that mostly uses only a small fraction of the features even though nearly as high accuracy is achievable using only the remaining features.  In the last part of the paper, the authors propose an ensemble obtained by adding a second network that is trained only on the features that are mostly not used by the first network, and report some experimental results that show performance superior to an ensemble of two independently trained networks, in particular when the features are corrupted by Gaussian noise.

**Strengths:**

The paper addresses one of the greatest current challenges in machine learning research, namely understanding the bias of gradient-based algorithms for training neural networks.  The proofs of the theoretical results are provided in the appendix, which also contains additional material on the experiments and a further discussion of related work.  The setup for the experiments is realistic, using a pretrained Resnet-50 network and some challenging datasets.  The paper is well written.

**Weaknesses:**

The connections between the theoretical and the experimental results are a little loose, e.g. it is not clear to what extent the projection of the data by P is linearly separable; some depictions and measurements of this are in Figure 8 and Table 6 in the appendix, however they are not very conclusive.  The paper by Galanti and Poggio, cited in Section 2 and twice in Appendix C, is v1 of arXiv:2206.05794, where v2 and v3 have two additional authors and a different title; also a submission with the same title and authors as the latter was rejected by ICLR 2023 (https://openreview.net/forum?id=N7Tv4aZ4Cyx), so I am not sure this paper is sound, even its v3.

**Questions:**

It seems to me that the literature on implicit bias of gradient descent is more related to the present paper than you suggest by reviewing it briefly and at the ends of Section 2 and Appendix C; after all, the work using which you prove the main theoretical result in the paper is from this literature (Chizat and Bach COLT 2020).  Related to this, you write in Appendix C that "for non-linear networks, the work related to low-rank simplicity bias is rather sparse", and in addition to Galanti and Poggio (discussed above in this review) only cite the empirical work of Huh et al.  You may want to refer to Phuong and Lampert ICLR 2021, Wang and Pilanci ICLR 2022, Sarussi et al. ICML 2021, Frei et al. arXiv:2210.07082 and ICLR 2023, Timor et al. ALT 2023, Boursier et al. NeurIPS 2022, and Chistikov et al. arXiv:2306.06479.

It would also be interesting to compare the present work with some of the most related literature on adversarial examples and adversarial robustness, such as Shamir et al. arXiv:2106.10151, Vardi et al. "Gradient Methods Provably Converge to Non-Robust Networks" NeurIPS 2022, Melamed et al. arXiv:2303.00783, Lv and Zhu ICLR 2022.

Minor comment: something went wrong in footnote 1?


**Limitations:**

The theoretical results are for infinite-width networks.  In connection with their transfer to finite widths, another submission to NeurIPS 2023 is cited (Anonymous "Feature-learning networks are consistent across widths at realistic scales") and included in the supplementary materials, however that work is empirical rather than theoretical.

---

> ### Author Rebuttal · Authors · 2023-08-09
>
> We thank the reviewer for their insightful comments. The specific concerns are addressed below:
>
> 1. **Linear separability of the dataset** - Since the embeddings on real datasets are not linearly separable, we plotted the decision boundaries as well as linear classification accuracies to qualitatively/quantitatively measure closeness to linear decision boundaries. Qualitatively, the results suggest that the initial network has a simpler boundary.
>
> 2. **More related works** - Thanks for pointing out the various related works. We will make sure to include these works in the next revision with a proper discussion about them.
>
> 3. **Infinite width networks** - Yes, the anonymous cited work does an empirical analysis of whether the training dynamics of finite width networks currently being used in practice have started to converge to the infinite width limit, and finds reasonable agreements.

---

> > ### Comment · Reviewer_gAuS · 2023-08-10
> >
> > Thank you for these responses.

---

### Official Review · Reviewer_aqCb · 2023-07-04

**Soundness:** 3 good
**Presentation:** 2 fair
**Contribution:** 3 good
**Rating:** 6
**Confidence:** 4

**Summary:**

This work characterizes of a type of simplicity bias (SB) in 1 hidden layer neural networks (NNs), termed as LD-SB. A model exhibits LD-SB when its predictions almost entirely depend on a low-dimensional projection of the input.

For the theoretical analyses, the authors consider NNs with infinite width, in both rich and lazy initialization regimes, and the independent features model. They show that when the data is linearly separable in one co-ordinate, the model almost exclusively relies on a single feature, even in the presence of several other complex features that are non-linearly separable.

The authors conduct experiments to verify that 1 hidden layer NNs exhibit LD-SB. They consider four real datasets, including Imagenette, Imagenet, Waterbird and employ various quantifiers to demonstrate LD-SB.

Finally, they propose a novel ensemble method called OrthoP. This method encourages feature learning by sequentially training NNs on projections of input data, by separating out directions already learned by previous NNs. The authors demonstrate that this ensemble method learns models that leads to improved robustness to Gaussian noise.

**Strengths:**

1. This work gives a precise characterization of LD-SB, and the quantifiers utilized in Section 4.4.1 to demonstrate LD-SB are well-motivated based on the definition of LD-SB (Def 1.1).

2. The paper as a whole is intriguing. It builds upon existing works and, although the individual contributions may not be highly significant, the ideas are interconnected effectively, providing valuable insights into the behavior of one hidden layer neural networks.

**Weaknesses:**

1. The statements regarding the contribution in relation to prior work on simplicity bias (line 13, line 78, and the related work paragraph 1), such as [1], overlook certain aspects and do not accurately reflect the scope of the contribution.

2. Definition 1.1 and its accompanying description require further clarification.

3. Section 5, which presents empirical validation of LD-SB on real datasets, as well as the proposed ensemble method OrthoP, could be strengthened by including validation on additional datasets and conducting comparisons with more methods, respectively.

4. The presentation of the paper should be refined. Some notation is non-standard or inconsistent, and the figures appear too small, which affects readability.

**Questions:**

$1.$ Contribution in relation to prior works on simplicity bias (SB):

Previous works on SB, such as [1], focus on a data setting where one feature is linearly separable with a smaller margin, alongside other features that are non-linearly separable. In that setting, using all the features leads to a predictor with a better margin.

However, this work states that there exists a unique max-margin predictor that solely relies on the linearly separable feature (theorem 4.1). Based on my understanding, the predictor that uses the non-linearly separable feature in this setting would have a smaller margin. This is also consistent with the experimental results in Fig. 4, which illustrate that an ensemble of $f$ and $f_{\text{proj}}$ has comparable or worse performance compared to using only $f$, when there is no Gaussian noise. Consequently, it appears reasonable to expect the neural network to learn the linear predictor.

In contrast, [1] and [2] demonstrate (empirically and theoretically, respectively) that 1 hidden layer NNs learn the max-margin linear predictor even when the global max-margin predictor is non-linear.

I suggest including a discussion to clarify this distinction in comparison to other works. This would help elucidate that the SB characterization examined in this work is slightly different, and this form of SB is not necessarily surprising or detrimental.

$2.$ Regarding Definition 1.1:

The definition of LD-SB does not explicitly mention that model $f$ obtains high accuracy on samples from distribution $D$. Lines 66-68 state that the third condition implies model $g$ obtains high accuracy, which may not always be true (e.g., if accuracy of $f$ is low). The definition implicitly assumes that $f$ has high accuracy, but this should be mentioned clearly.

$3.$ Section 5: Validation on additional datasets and comparison of OrthoP with other methods:

- The empirical validation of LD-SB in 1 hidden layer NNs on real datasets can be made more extensive, by including more datasets, such as the CelebA dataset or variants of the Imagenet dataset considered in [3].

- Consider comparing OrthoP with other rich feature learning methods, such as gradient starvation [4], etc.

- Fig. 3 should include results for Imagenet dataset, and the MNIST-CIFAR results from the Appendix (Fig. 7) can be incorporated into Fig. 4.

$4.$ Presentation:

- Regarding notation:

    - The notation for indexing two datapoints seems inconsistent (compare line 56 and line 63).

    - Consider using standard symbols for the set of real numbers (line 145), matrices (line 147: use $\mathbf{W}$ instead of $w$), vectors, etc. Also, moving lines 152-154 to the beginning of Section 3.1 could enhance clarity.

    - The notation $\bar{(\cdot)}$ is not defined. $f(x)$ and $f((\bar{w},\bar{b},\bar{a}),x)$ are used interchangeably, but without stating that these are normalized parameters, due to which the definition of margin is not clear.

- Figures:

    The quality of the figures should be improved. In most cases, the axis labels and legends are too small and difficult to read. In Fig. 4, the blue and green colors are hard to distinguish, and thicker lines in Figs. 3 and 4 could enhance comprehension.

- Other points:

    - Consider relocating Table 1 to page 8 so that those values can be compared with the values in Table 2.

    - The Waterbirds dataset is referred to as waterbirds-landbirds dataset in some instances.

    - Consider adding related work [5], which empirically demonstrates that using SGD with a large step size encourages neural networks to learn sparse solutions.

References:

[1] Shah, H., Tamuly, K., Raghunathan, A., Jain, P., and Netrapalli, P. The pitfalls of simplicity bias in neural networks. Advances in Neural Information Processing Systems, 33:9573–9585, 2020.

[2] Lyu, K., Li, Z., Wang, R., and Arora, S. Gradient descent on two-layer nets: Margin maximization and simplicity bias. Advances in Neural Information Processing Systems, 34:12978–12991, 2021.

[3] P. Kirichenko, P. Izmailov, and A. G. Wilson. Last layer re-training is sufficient for robustness to spurious correlations. arXiv preprint arXiv:2204.02937, 2022.

[4] Pezeshki, M., Kaba, O., Bengio, Y., Courville, A. C., Precup, D., and Lajoie, G. Gradient starvation: A learning proclivity in neural networks. Advances in Neural Information Processing Systems, 34:1256–1272, 2021.

[5] Andriushchenko, M., Varre, A., Pillaud-Vivien, L., and Flammarion, N. SGD with large step sizes learns sparse features. In ICML, 2023.

**Limitations:**

[+] The authors have provided the code and experimental details to support reproducibility.

---

> ### Author Rebuttal · Authors · 2023-08-09
>
> We sincerely thank the reviewer for their valuable comments and suggestions, and are happy to know that the reviewer finds our paper intriguing and insightful. The specific concerns are addressed below:
>
> 1. **Contribution in relation to prior works on simplicity bias (SB)** - Thanks for pointing out this issue. The subtlety here is that there are two different notions of margin considered in different papers.
>
>     a. *Max input space margin* refers to maximum distance between the classification boundary and the data points in the input space. [1]  uses this notion.
>
>     b. *Max F1 margin* refers to minimizing L2 norm of the parameters. Both [2] and (Chizat and Bach, 2020) use this notion. (Chizat and Bach, 2020) shows that asymptotically, neural networks with infinite width converge to this classifier always, while [2] discusses early training dynamics and finite width networks.
>
>     Our key contribution is to show that these two notions can be very different. In fact, our IFM datasets have the property that the max input space margin classifiers are nonlinear while the max F1 margin classifier is linear. Consequently, neural networks which converge to the max F1 margin classifier suffer from simplicity bias in the context of max input space margin.
>
> 2. **Regarding Definition 1.1** - Thanks for the catch. Yes, we will mention that we are assuming $f$ to have high accuracy.
>
> 3. **Validation on additional datasets and comparison of OrthoP with other methods** - Regarding empirical validation, we have obtained results for comparison to an additional method - Evading-SB [3]. We have added this to the one page pdf. Our method performs either comparable or slightly better than this method. We will also move results for Imagenet and for MNIST-CIFAR from the appendix to the main paper.
>
> 4. **Presentation** - Thanks for the detailed suggestions on presentation. We will improve the presentation and increase the size of figures as per your suggestions.
>
> [1] Shah, H., Tamuly, K., Raghunathan, A., Jain, P., and Netrapalli, P. The pitfalls of simplicity bias in neural networks. Advances in Neural Information Processing Systems, 33:9573–9585, 2020.
>
> [2] Lyu, K., Li, Z., Wang, R., and Arora, S. Gradient descent on two-layer nets: Margin maximization and simplicity bias. Advances in Neural Information Processing Systems, 34:12978–12991, 2021.
>
> [3] - Teney, D., Abbasnejad, E., Lucey, S., and van den Hengel, A. Evading the simplicity bias: Training a diverse set of models discovers solutions with superior ood generalization. In IEEE/CVF Conference on Computer Vision and Pattern Recognition, pp. 16761–16772, 2022

---

> > ### Comment · Reviewer_aqCb · 2023-08-12
> >
> > Thank you for these responses.

---

### Official Review · Reviewer_PpnG · 2023-07-20

**Soundness:** 3 good
**Presentation:** 1 poor
**Contribution:** 2 fair
**Rating:** 6
**Confidence:** 3

**Summary:**

The paper studies the simplicity bias (SB) for one hidden layer neural networks. Specifically, it I) provides a formal definition of SB for 1-hidden layer NN, II) proves that simplicity bias takes place when a infinite-width 1-hidden layers NNs are trained on linearly separable datasets with a gradient descent algorithm, and III) demonstrates SB on real datasets, motivating a new ensemble approach to encourages diversity in models. The paper contributes on both the theory behind the why simplicity bias happens, and the practices of how to avoid SB in real datasets.

**Strengths:**

1. The research targets at simplicity bias, a crucial problem facing neural networks.

2. The paper defines and proves existence of a more general SB on a more general type of datasets than the existing work, significantly advancing the theory of SB.

3. The paper makes a connection between theory and practice by empirically studying the Low-dimensional SB on five real datasets.

**Weaknesses:**

1. The contributions of the paper are poorly presented in the introduction. The definition 1.1 is hard to understand due to the missing notations (e.g. x(1), y(1), P()).  It should be moved to the later chapters. Also, the clarification on why the focus is set on infinite-width regime is not convincing, especially the practical standpoint.

2. The preliminaries are a bit too technical for the general audience. For instance, equations 1), 2) and 3) could be explained with plain and easy-to-understand language.

3. It is claimed that the achieved result is a generalization of Shah et al. (2020)'s work. However, it is not discussed which special case of this work does Shah's work fall into.

4.  The empirical validation part is not as promising as the theoretical part. For instance, the laze regime seems to be a general regime for LD-SB instead of a specific strategy for the lazy initialization of weight; the Rich regime is not comprehensively evaluated by compared with the same two-step approach but other ways of constructing P (and P_ortho).

**Questions:**

1. Could you please describe how the theorems will fail if the 1-hidden layer NN is not of infinite width?

2. Could you please further elaborate how Shah et al.'s work fits in your findings as a special case?

3. Why not empirically validate the LD-SB in other artificially created data in the IFM than the ones used in Shah et al. (2020)'s work?

**Limitations:**

The authors have adequately addressed the limitations.

---

> ### Author Rebuttal · Authors · 2023-08-09
>
> We thank the reviewer for their valuable feedback. The specific concerns are addressed below:
>
> 1. **Presentation of Introduction** - Thanks for the suggestion about presentation in Definition 1.1. We will add exposition and unpack it for readers’ convenience. We would like to clarify that the focus on infinite width setting is primarily because of theoretical tractability. As we mention in lines 97-100, in the feature learning regime, recent works show that finite width neural networks are well approximated by their infinite width counterparts and our empirical results confirm that our theoretical results are valid for finite width networks used in practice.
>
> 2. **Preliminaries** - We introduced the technical preliminaries to present our results rigorously. We are happy to add more exposition, but are constrained by the page limits.
>
> 3. **Relation to Shah et.al** - Thanks for the question. The dataset used in Shah et al, is a particular instance of the IFM dataset, where one coordinate is linearly separable and other coordinates are separable using 2 linear pieces. Instead, our proof works for any linearly separable dataset within the IFM family. We will include this in the updated draft.
>
> 4. **Lazy regime and Rich regime** - Sorry, we couldn’t completely understand the comment of the reviewer. Lazy regime training is indeed dependent on the initialization, and we showcase that it still exhibits LD-SB even when the weights themselves are not low rank. For rich regime, since the weights themselves are low rank, they clearly specify the low dimensional subspace used by the model. Thus, we consider it to be an appropriate choice for P.
>
> 5. **Infinite width** - Our theoretical results rely on the characterization provided in Theorem 3.1 by Chizat and Bach (2020). Their proof works only for the infinite width, and hence our theoretical results also hold only for the infinite width. However, our empirical results clearly show that our theory is also valid for the finite width models used in practice.
>
> 6. **Other synthetic datasets** - Given that earlier works such as Shah et al. already demonstrated in specialized, synthetic datasets within IFM, the focus of our work is to extend the generality of these results both theoretically and empirically. So, empirically, we focused on real world datasets. We can include results on other synthetic datasets in the appendix, but are not sure if it will teach us anything new.
>
> We hope that we addressed the reviewers questions and concerns and would appreciate if the reviewer would update their score accordingly.

---

> > ### Comment · Reviewer_PpnG · 2023-08-18
> >
> > I would like to thank the authors for their response.  The majority of my concerns are well addressed. Therefore, I will change the rating from 4 to 6.
> >
> > Just a further clarification on my comment on the lazy and rich regimes: in the experiment, one specific strategy of both regimes are used for deriving the P; I think it could be beneficial to examine the performance of other lazy and rich regimes of constructing P, because it will better demonstrate that LD-SB holds on real dataset, and also gives rise to better OrthoP classifiers. Correct me if I am wrong.

---

> > > ### Author Response · Authors · 2023-08-21
> > >
> > > Thanks for updating the score. Regarding the choice of $P$ in rich and lazy regimes, it could be obtained using different strategies as well. However, for demonstrating LD-SB, we need to find one such P, which our strategy finds. Moreover, for rich regime, we do think that this is indeed the *right* choice, as can be seen by Theorem 4.1, where the top singular vectors of the weight matrix highlight the linearly separable coordinate within IFM.

---

### Author Rebuttal · Authors · 2023-08-09

We thank the reviewers for their insightful comments. Based on the suggestions of the reviewers, we conducted a comparison experiment with another diverse training method - Evading SB [1]. The results are attached in the one page pdf and summarized below:

1. Our method generally performs comparable or slightly better than Evading-SB in terms of class conditioned logit correlation and mistake diversity.
2. In terms of gaussian robustness, both methods are comparable, with our method slightly outperforming on ImageNet and Evading-SB slightly outperforming on MNIST-CIFAR.

[1] Teney et al. - Evading the Simplicity Bias: Training a Diverse Set of Models Discovers Solutions with Superior OOD Generalization, CVPR, 2022

---

### Decision · Program_Chairs · 2023-09-21

**Decision:**

Accept (poster)

**Comment:**

In this paper, the authors introduce a definition of simplicity bias of a model (in terms of its being a function of a low-dimensional projection of input data.  The authors rigorously prove, in an infinite-width 1-hidden-layer neural network regime, that when data is linearly separable in a single dimension, the model will find that dimension, even if the resulting networks is less robust than other potential models.  The authors illustrate their notion of simplicity bias on networks trained on real datasets such as ImageNet.  Finally, the authors introduce an ensemble method that yields models that are more robust to Gaussian noise.  The main strength of the paper is that it provides an advance to the theory of simplicity bias, which is a central challenge of the deep learning field.   There were some concerns that the theory and experimental sections were not as tightly related as they could be, but given the difficulty of theoretical advances in this area, those differences are to be expected.